# EFFICIENT REDUNDANCY-FREE GRAPH NETWORKS: HIGHER EXPRESSIVENESS AND LESS OVER-SQUASHING

## ABSTRACT

Message Passing Neural Networks (MPNNs) are effective at learning graph structures. However, their message passing mechanism introduces redundancy, which limits expressiveness and leads to over-squashing. Previous studies have addressed the redundancy problem, but often at the cost of increased complexity. Improving expressiveness and addressing over-squashing remain major concerns in MPNN research. This study explores the nature of message passing redundancy and presents efficient solutions using two surrogate structures: Directed Line Graph (DLG) and Directed Acyclic Line Graph (DALG). Subsequently, we propose a family of models: 1) Directed Line Graph Network (DLGN), using DLG, achieves redundancy-free message passing for graphs with a minimum cycle size of $L$ when composed of $L$ layers. 2) Efficient Redundancy-Free Graph Network (ERFGN), using DALG, achieves perfectly redundancy-free message passing and, under certain conditions, the expressiveness of arbitrary subtrees. 3) Extending ERFGN with cycle modeling, ERFGN$^\circ$ expresses each subgraph with a cycle. 4) Performing **Glo**bal **Att**ention of **N**odes and **C**ycles(GloAttNC), ERFGN$^\circ$+GloAttNC achieves a high expressiveness of subgraphs consisting of subtrees and subcycles. The efficiency and effectiveness of these models in improving expressiveness and mitigating over-squashing are analyzed theoretically. Empirical results on realistic datasets validate the proposed methods.

## 1 INTRODUCTION

Message Passing Neural Networks (MPNNs), the cornerstone branch of Graph Neural Networks (GNNs), have shown strong performance in graph learning (Waikhom & Patgiri, 2021; Xia et al., 2021; Khoshraftar & An, 2022; Yang et al., 2023; Dwivedi et al., 2023). In typical MPNNs (Gilmer et al., 2017; Kipf & Welling, 2017; Hamilton et al., 2017; Velickovic et al., 2018; Xu et al., 2019), a node representation is computed by aggregating messages from the node itself and its neighbors, where the input graph is transformed into a *surrogate graph* consisting of bi-directed edges and self-loops, as shown in Fig. 1 (a). Typical MPNNs implicitly build a *message passing graph*, as depicted in Fig. 1 (b), to pass messages and compute node representations. The *expressiveness* of GNNs quantifies their ability to identify various non-isomorphic (i.e., topologically mismatched) (sub)graphs (Xu et al., 2019; Morris et al., 2019). To effectively measure graph similarity and accurately predict graph properties, it is crucial to express a wide range of non-isomorphic subgraphs (Tsitsulin et al., 2018). Conversely, an approach that effectively separates non-isomorphic graphs but fails to express isomorphic subgraphs within them may encounter the overfitting issue. However, MPNNs face challenges in generating identifiable representations for non-isomorphic subgraphs. These challenges arise from two main issues: *message confusion* and *over-squashing* (Alon & Yahav, 2021; Topping et al., 2022; Di Giovanni et al., 2023). Additionally, Chen et al. (2022) revealed that MPNNs also suffer from the *message passing redundancy* issue, which causes the *message confusion* issue and contributes significantly to the *over-squashing* issue.

The phenomenon of *message passing redundancy* can be characterised from two aspects: (i) the existence of redundant subpaths within a message passing path, and (ii) the existence of multiple message passing paths generated from one source path. For instance, a path $b{\to}a$ can generate multiple message passing paths, e.g., $a{\to}b{\to}a$, $b{\to}b{\to}a$ and $b{\to}a{\to}a$ as shown in Fig. 1(b), because the self-loops (e.g., $a{\to}a$ and $b{\to}b$) and backtracking edges (e.g., $a{\to}b$) always generate redundant

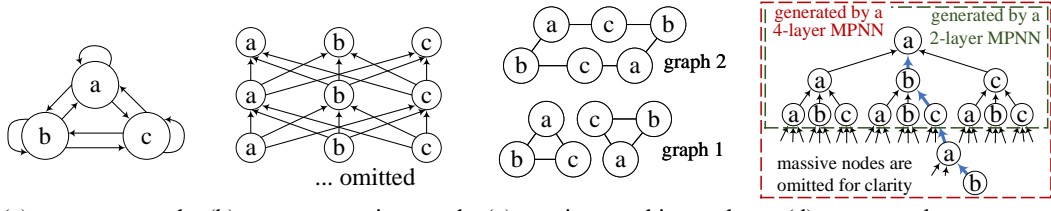

Figure 1: Features of typical message passing paradigm.

subpaths. In addition, a closed path $b\rightarrow a\rightarrow c\rightarrow b$ in graph 1 or a path $b\rightarrow a\rightarrow c\rightarrow b\rightarrow a$ in graph 2 shown in Fig.1(c) both can generate the message passing path $b\rightarrow a\rightarrow c\rightarrow b\rightarrow a$ shown in blue in Fig.1(d). These examples imply that *message passing redundancy* causes challenges in determining the source path of a given message passing path, known as the *message confusion* problem. As a result, typical MPNNs face challenges in solving tasks involving inter-node distances. Moreover, they can generate identical message passing graphs for non-isomorphic subgraphs, as exemplified by the message passing graph in Fig. 1(d), which is generated for the two non-isomorphic graphs depicted in Fig.1(c). This issue leads to indistinguishable representations being generated for non-isomorphic (sub)graphs, further limiting the expressiveness of typical MPNNs. The *over-squashing* phenomenon describes the distortion of messages that occurs when a node's exponentially expanding neighborhood information is compressed or 'squashed' into vectors of limited size (Alon & Yahav, 2021; Topping et al., 2022). For instance, any node labeled as $a$ in Fig. 1(c) receives messages from its exponentially growing neighborhoods shown in Fig. 1(d). Chen et al. (2022) showed that a source path can be transformed into its derived message passing paths by adding redundant subpaths. This suggests that typical MPNNs always generate many more equal-length message passing paths from short paths than from long paths, because short paths can be extended with more redundant subpaths. Consequently, typical MPNNs may lose sensitivity in capturing long paths and large subgraphs. Hence, *message passing redundancy* significantly contributes to the *over-squashing* issue.

Redundancy-free message passing is undoubtedly crucial. However, previous studies in this area have failed to achieve this goal efficiently and perfectly. A pioneering study, RFGNN (Chen et al., 2022), conducts message passing on a surrogate structure called TPTs, which consist of extended paths (epaths) extracted from source graphs, where an epath can represent a cycle. The use of TPTs ensures that only one message passing path captures an epath, making RFGNN redundancy free. Another study, PathNNs (Michel et al., 2023), also extracts path trees to achieve redundancy-free computation. However, both RFGNN and PathNNs suffer from computational and memory complexity due to the exponential growth in the size of the path trees. This limits their scalability and applicability in practical scenarios. In addition, SPAGAN (Yang et al., 2019) introduces shortest-path-based attention, which improves efficiency but sacrifices subtree topological information. Furthermore, there are other studies that focus on eliminating backtracking, such as (Mahé et al., 2004; Chen et al., 2018). However, these methods still suffer from redundancy caused by subcycles.

We carefully study the causes of message passing redundancy. Surrogate structures generated by typical MPNNs contain massive subcycles, including *self-loops*, *backtrackings*, and *native cycles*, as shown in Fig. 1(a). We find that when messages repeatedly traverse subcycles within surrogate graphs, MPNNs generate redundant message passing. Therefore, to achieve efficient redundancy-free message passing, we propose solutions based on surrogate structures that eliminate cycles. First, we utilize a surrogate structure called Directed Line Graph (DLG). A DLG excludes self-loops and backtracking edges by representing directed edges [1] of a source graph as nodes and limiting node connections to non-backtracking pairs. Our DLG structure not only provides efficiency and reversibility but also enables the representation of source subtrees using DLGs, which makes our DLG structure being different to the directed line graphs used in previous work (Mahé et al., 2004) and (Chen et al., 2018) (detailed in Appendix A). However, it retains native cycles. Therefore, we propose a method to convert source graphs into cycle-free DLGs, a.k.a Directed Acyclic Line Graphs (DALGs). To ensure efficiency, this method extracts path trees from a cyclic subgraph rather than the entire graph. It achieves this by decomposing the source graph into two subgraphs: cyclic and

---

[1]Undirected edges are treated as bi-directed in this study.

acyclic. A sub-DALG is then constructed by extracting path trees from the cyclic subgraph, while another sub-DALG is created from the acyclic subgraph using the DLG conversion. Finally, the two sub-DALGs are merged together. The DALG conversion provides reversibility, and efficiency over than path tree extraction. The DLG and DALG introduce two corresponding models: Directed Line Graph Network (DLGN) and Efficient Redundancy-Free Graph Network (ERFGN). DLGN with $L$ layers achieves efficient redundancy-free message passing by avoiding message repetition within subcycles of size $L$ or greater. ERFGN achieves both efficiency and perfection in redundancy-free message passing due to the limited size and cycle-free nature of DALGs. As well as improving efficiency, we are also improving expressiveness. ERFGN effectively express subtrees. By extending ERFGN with cycle modeling, we introduce ERFGN$^\circ$ to express any connected subgraph with a subcycle. We also propose a **Glo**bal **Att**ention module to model **N**ode and **C**ycle interactions (called GloAttNC). It allows ERFGN$^\circ$+GloAttNC to express subgraphs consisting of subcycles and subtrees, and to generate identifiable representations for nodes and cycles.

Our contributions of this work could be summarized as follows:

- Our investigation reveals that the *message passing redundancy* arises from the repetitive traversal of messages along cyclic structures within surrogate graphs.
- To achieve efficient redundancy-free message passing, we propose two solutions using surrogate structures that eliminate cycles. First, we propose DLGN, which utilizes DLG, a surrogate structure that excludes self-loops and backtrackings. When DLGN is composed of $L$ layers, it achieves redundancy-free message passing for graphs with a minimum cycle size of $L$. Second, we introduce ERFGN, which employs cycle-free surrogate graphs, known as DALGs. We also present a method for efficiently converting cyclic graphs into DALGs, which extracts path trees from cyclic subgraphs rather than from entire graphs. ERFGN achieves perfect redundancy-free message passing and, under certain conditions, enables the expressiveness of arbitrary subtrees.
- To extend ERFGN's ability to capture higher-order graph structures, we introduce ERFGN$^\circ$, which includes a cycle modeling module to express any subgraph with a cycle. Additionally, we propose GloAttNC, a methodology that models global attentions between nodes and cycles. By incorporating GloAttNC, ERFGN$^\circ$+GloAttNC can effectively generate identifiable representations for nodes and cycles.
- Empirical results strongly support the efficiency and high performance of our models.

## 2 METHODOLOGY

In this section, we clarify the research problem, present the DLG conversion and DALG conversion, introduce our proposed models, and conduct a comprehensive evaluation of their expressiveness and complexity. The detailed proofs can be found in Appendix C.

### 2.1 PROBLEM DEFINITION

Let $\mathcal{G} = (V, E, C)$ be a graph. A directed edge $e = (u, v) \in E$ connects node $u$ to node $v$, where $u, v \in V$. $c \in C$ is a chordless cycle (in which no edge outside the cycle connects two nodes of the cycle (Uno & Satoh, 2014)). Each node $u$ and each edge $e$ could have attributes, denoted as $X_u$ and $X_e$, respectively. This study focuses on graph-level learning tasks: First, a model learns a node representation $Z_u^{\cdot}$ for each node $u \in V$. Then, the model learns a graph representation $Z_{\mathcal{G}}^*$ based on the node representations of graph $\mathcal{G}$, which is used to predict the properties of graph $\mathcal{G}$. [2]

### 2.2 DLG CONVERSION

**Definition 1** (Directed Line Graph (DLG)). *The DLG $\mathcal{D}$ of a graph $\mathcal{G}$ is a graph such that (i) each node of $\mathcal{D}$ represents a directed edge of $\mathcal{G}$; and (ii) two nodes in $\mathcal{D}$ are adjacent if and only if their source edges connect but do not form a backtracking.[3] Particularly, an acyclic DLG is called a Directed Acyclic Line Graph (DALG). A DLG node is denoted as $v{\rightarrow}u$ with the source edge $(v, u)$.*

---

[2] Representations and functions of nodes are indicated with a superscript $\cdot$, cycles with $\circ$, and graphs with $*$.

[3] The concept of 'Directed Line Graph' is developed based on the 'Line Graph'.

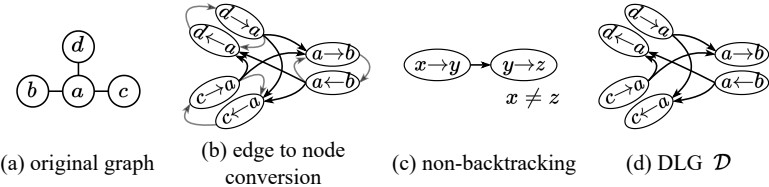

Figure 2: Eliminating self-loops and backtrackings with DLGs.

The DLG conversion is illustrated in Fig. 2: First, the original directed edges are represented as DLG nodes, as shown in Fig. 2(b). Second, DLG node connections are limited to non-backtracking node pairs. A non-backtracking node pair is shown in Fig. 2(c). The resulting DLG is shown in Fig. 2(d). At last, for each node $u \in \mathcal{G}$, a set $\mathbf{D}(u) = \{v \to u \mid (v, u) \in \mathcal{G}\}$ is created, which contains all the DLG nodes generated from incoming edges of node $u$, making these DLG nodes virtually connected to node $u$. It is essential to highlight that this method transforms acyclic graphs into DALGs.

## 2.3 DALG CONVERSION

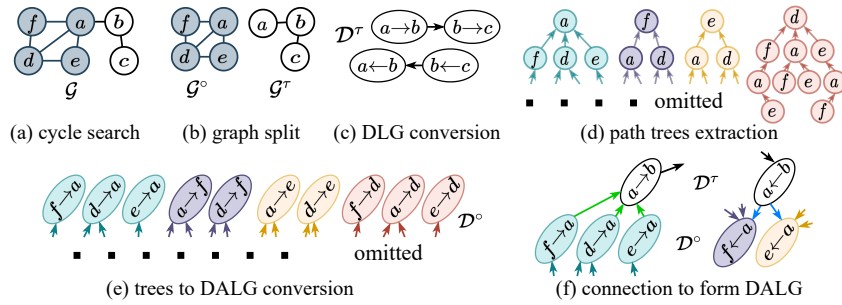

Figure 3: Steps of DALG Conversion

The steps of DALG conversion is shown in Fig. 3 and summarized as follows:

(a) Chordless cycles present in the input graph are extracted. If no cycles are found, the input graph is converted directly into a DALG using the DLG conversion method.

(b) If cycles are detected, the input graph is divided into two subgraphs, i.e., the cyclic subgraph consisting of all subcycles, denoted as $\mathcal{G}^\circ$, and the complementary acyclic subgraph, denoted as $\mathcal{G}^\tau$. The two subgraphs may have joint nodes, e.g., the node $a$ shown in Fig. 3(b).

(c) $\mathcal{G}^\tau$ is converted into a sub-DALG, denoted as $\mathcal{D}^\tau$, using the DLG conversion.

(d) Path trees are extracted from each node in $\mathcal{G}^\circ$. Path trees connect a node to its parent node if and only if the two nodes are adjacent in $\mathcal{G}^\circ$. And each path within path trees contains unique nodes.

(e) Path trees are converted into a sub-DALG, called $\mathcal{G}^\circ$, by representing their edges as DALG nodes. And, a set $\mathcal{E}$ is created including all DALG node generated from the root edges of path trees.

(f) The sub-DALGs $\mathcal{D}^\tau$ and $\mathcal{D}^\circ$ are merged to construct a complete DALG. If the cyclic and acyclic subgraphs have joint nodes, edges are placed between these two sub-DALGs:

  (i) An edge (colored green) is added from a node in $\mathcal{D}^\circ$ to a node in $\mathcal{D}^\tau$ if and only if the source edges of these nodes are adjacent and the $\mathcal{D}^\circ$ node originates from a root edge of a path tree. By adding such edges, every path from $\mathcal{G}^\circ$ to $\mathcal{G}^\tau$ is present in the DALG. Note that nodes in $\mathcal{D}^\circ$ whose source edges are non-root edges are not connected to nodes in $\mathcal{D}^\tau$ because paths ending at non-root edges are duplicated in paths ending at root edges.

  (ii) An edge (colored blue) is placed from a $\mathcal{D}^\tau$ node to a $\mathcal{D}^\circ$ node if and only if the source edges of these DALG nodes are adjacent, so that all paths from $\mathcal{G}^\tau$ to $\mathcal{G}^\circ$ present in the DALG.

(g) In addition, for each node $u$, a set $\mathbf{D}(u) = \{v \to u \mid (v, u) \in \mathcal{G}^\tau \cup (v, u) \in \mathcal{E}\}$ is created, which contains all DALG nodes whose source edges pointing to node $u$ and generated from $\mathcal{G}^\tau$ or the root edges of path trees, making DALG nodes in $\mathbf{D}(u)$ virtually connected to node $u$. And, for each cycle $c$, a set $\mathbf{C}(c) = \{v \to u \mid (v, u) \in c \cap (v, u) \in \mathcal{E}\}$ is created, which contains all DALG

nodes whose source edges belong to cycle $c$ and are from root edges of trees, making DALG nodes in $\mathbf{C}(c)$ virtually connected to cycle $c$. DALG nodes that are generated from non-root edges of path trees are excluded in both $\mathbf{D}(u)$ and $\mathbf{C}(u)$ to avoid duplicate messages.

## 2.4 DLGN AND ERFGN

The DLG and DALG introduce two corresponding models: DLGN and ERFGN. Both first perform message passing on surrogate graphs to update the node representations of surrogate graphs, then update the source node representations, and finally update the graph representations. Since DALGs are special DLGs, the message passing schemes applied to DLGs and DALGs are the same. For a given graph $\mathcal{G} = (V, E, C)$ and its derived DLG $\mathcal{D}$, an initial representation $H_{\mathbf{u}}^{(0)}$ is encoded for each DLG node $\mathbf{u} \in \mathcal{D}$ using an update function $\Psi$ as follows:

$$H_{\mathbf{u}}^{(0)} = \Psi\left(X_u, X_{u,v}, X_v\right), \tag{1}$$

where node $\mathbf{u}$ is generated from the directed edge $(u, v) \in \mathcal{G}$, node or edge attributes could be empty. Then, with $H_{\mathbf{u}}^{(l)}$ denoting the representation of node $\mathbf{u}$ in layer $l$, $H_{\mathbf{u}}^{(l)}$ is updated as follows:

$$H_{\mathbf{u}}^{(l)} = \Psi^{(l)}\left(H_{\mathbf{u}}^{(0)}, \bigcup{}^{(l)}\left(\left\{\left\{H_{\mathbf{v}}^{(l-1)} \;\middle|\; \mathbf{v} \in \mathcal{N}_{in}(\mathbf{u})\right\}\right\}\right)\right), \tag{2}$$

where $\mathcal{N}_{in}(\mathbf{u})$ is the set of incoming DLG nodes of node $\mathbf{u}$, and $\bigcup^{(l)}$ is a message aggregator. The update function $\Phi^{(l)}$ takes $H_{\mathbf{u}}^{(0)}$, not $H_{\mathbf{u}}^{(l-1)}$ as in typical MPNNs, in order to avoid self-loops.

Subsequently, DLGN and ERFGN employ a similar scheme to update the source node representations. The representation $Z_u^{\cdot}$ of node $u$ is computed using an update function $\Psi^{\cdot}$ and a message aggregator $\bigcup^{\cdot}$ in the following manner:

$$Z_u^{\cdot} = \Psi^{\cdot}\left(\bigcup{}^{\cdot}\left(\left\{\left\{H_{\mathbf{u}}^{\cdot} \;\middle|\; \mathbf{u} \in \mathbf{D}(u)\right\}\right\}\right)\right), \tag{3}$$

where $H_{\mathbf{u}}^{\cdot}$ could be $H_{\mathbf{u}}^{(L)}$ or $\left[H_{\mathbf{u}}^{(1)} \parallel H_{\mathbf{u}}^{(2)} \parallel \cdots \parallel H_{\mathbf{u}}^{(L)}\right]$. The concatenation of the node representations across layers, a.k.a Jump Knowledge (Xu et al., 2018), could help to incorporate graph features at different scales. DLGN and ERFGN differ in the definition of $\mathbf{D}(u)$: DLGN uses a $\mathbf{D}(u)$ including all DLG nodes whose source edges point to node $u$, while ERFGN uses a $\mathbf{D}(u)$ that excludes DALG nodes generated from non-root edges of path trees to avoid message passing redundancy.

Lastly, our models use a graph readout $\bigcup^{*}$ to collect the representations of all nodes of graph $\mathcal{G}$, and use an update function $\Psi^{*}$ to compute the representation $Z_{\mathcal{G}}^{*}$ of graph $\mathcal{G}$ as follows:

$$Z_{\mathcal{G}}^{*} = \Psi^{*}\left(\bigcup{}^{*}\left(\left\{\left\{Z_u^{\cdot} \;\middle|\; u \in V\right\}\right\}\right)\right). \tag{4}$$

All update functions are trainable functions. Additionally, the message aggregators $\bigcup^{(l)}$ and $\bigcup^{\cdot}$, and the graph readout $\bigcup^{*}$ must be permutation-invariant and injective functions on multisets, such as summation, where a multiset is a set that allows repeated elements.

## 2.5 CYCLE MODELING AND GLOBAL ATTENTION

To achieve higher expressiveness, we propose ERFGN$^{\circ}$, which explicitly models cycle-based subgraphs by adding a cycle modeling module to ERFGN. Specifically, ERFGN$^{\circ}$ updates the representation of each cycle $c \in C$ using a message aggregator $\bigcup^{\circ}$ and an update function $\Phi^{\circ}$ as follows:

$$Z_c^{\circ} = \Phi^{\circ}\left(\bigcup{}^{\circ}\left(\left\{\left\{H_{\mathbf{u}}^{(L)} \;\middle|\; \mathbf{u} \in \mathbf{C}(c)\right\}\right\}\right)\right). \tag{5}$$

Subsequently, ERFGN$^{\circ}$ computes the graph representation $Z_{\mathcal{G}}^{*}$ as follows:

$$Z_{\mathcal{G}}^{*} = \Phi^{*}\left(\bigcup{}^{*}\left(\left\{\left\{Z_u^{\cdot} \;\middle|\; u \in V\right\}\right\}\right), \bigcup{}^{*}\left(\left\{\left\{Z_c^{\circ} \;\middle|\; c \in C\right\}\right\}\right)\right). \tag{6}$$

To further extend the capabilities of ERFGN$^{\circ}$, we introduce GloAttNC, a global attention module which incorporates node-to-cycle, cycle-to-node, and cycle-to-cycle global interactions, in addition to the existing node-to-node interactions found in graph transformers (Kreuzer et al., 2021). It

employs two schemes, a *1-to-N scheme* and a *1-to-1 scheme*, to cater to different priorities. The *1-to-N scheme* prioritizes efficiency but may sacrifice some details. It updates the global interactions between a node $u$ to all nodes ($Z_{u,V}^{\cdot\cdot}$), between a node $u$ to all cycles ($Z_{u,C}^{\cdot\circ}$), between a cycle $c$ to all nodes ($Z_{c,V}^{\circ\cdot}$), and between a cycle $c$ to all cycles ($Z_{c,C}^{\circ\circ}$) using Eq. 7. Conversely, the *1-to-1 scheme* preserves details to potentially enhance accuracy but comes with increased computational overhead. It updates these global interactions using Eq. 8.

$$
\begin{aligned}
Z_{u,V}^{\cdot\cdot} &= \Upsilon^{\cdot\cdot}\left(Z_u^{\cdot}, \bigcup\left(\left\{\left\{Z_v^{\cdot} \mid v \in V\right\}\right\}\right)\right), & Z_{u,C}^{\cdot\circ} &= \Upsilon^{\cdot\circ}\left(Z_u^{\cdot}, \bigcup\left(\left\{\left\{Z_c^{\circ} \mid c \in C\right\}\right\}\right)\right), \\
Z_{c,V}^{\circ\cdot} &= \Upsilon^{\circ\cdot}\left(Z_c^{\circ}, \bigcup\left(\left\{\left\{Z_v^{\cdot} \mid v \in V\right\}\right\}\right)\right), & Z_{c,C}^{\circ\circ} &= \Upsilon^{\circ\circ}\left(Z_c^{\circ}, \bigcup\left(\left\{\left\{Z_{c'}^{\circ} \mid c' \in C\right\}\right\}\right)\right).
\end{aligned}
\tag{7}
$$

$$
\begin{aligned}
Z_{u,V}^{\cdot\cdot} &= \bigcup\left(\left\{\left\{\Upsilon^{\cdot\cdot}\left(Z_u^{\cdot}, Z_v^{\cdot}\right) \mid v \in V\right\}\right\}\right), & Z_{u,C}^{\cdot\circ} &= \bigcup\left(\left\{\left\{\Upsilon^{\cdot\circ}\left(Z_u^{\cdot}, Z_c^{\circ}\right) \mid c \in C\right\}\right\}\right), \\
Z_{c,V}^{\circ\cdot} &= \bigcup\left(\left\{\left\{\Upsilon^{\circ\cdot}\left(Z_c^{\circ}, Z_v^{\cdot}\right) \mid v \in V\right\}\right\}\right), & Z_{c,C}^{\circ\circ} &= \bigcup\left(\left\{\left\{\Upsilon^{\circ\circ}\left(Z_c^{\circ}, Z_{c'}^{\circ}\right) \mid c' \in C\right\}\right\}\right).
\end{aligned}
\tag{8}
$$

Subsequently, the representations of nodes, cycles and graphs are computed sequentially:

$$
\begin{aligned}
\mathbf{Z}_u^{\cdot} &= \Upsilon^{\cdot}\left(Z_u^{\cdot}, Z_{u,V}^{\cdot\cdot}, Z_{u,C}^{\cdot\circ}\right), \quad \mathbf{Z}_c^{\circ} = \Upsilon^{\circ}\left(Z_c^{\circ}, Z_{c,V}^{\circ\cdot}, Z_{c,C}^{\circ\circ}\right) \\
Z_{\mathcal{G}}^{*} &= \Upsilon^{*}\left(\bigcup\left(\left\{\mathbf{Z}_u^{\cdot} \mid u \in V\right\}\right), \bigcup\left(\left\{\mathbf{Z}_c^{\circ} \mid c \in C\right\}\right)\right).
\end{aligned}
\tag{9}
$$

Note that $\bigcup$ is a message aggregator, and $\Upsilon^{\cdot\cdot}$, $\Upsilon^{\cdot\circ}$, $\Upsilon^{\circ\cdot}$, $\Upsilon^{\circ\circ}$, $\Upsilon^{\cdot}$, $\Upsilon^{\circ}$, and $\Upsilon^{*}$ are trainable functions.

### 2.6 HIGHER EXPRESSIVENESS

**Lemma 1.** *All subtrees in a graph are represented as subtrees in the corresponding DLG. And the conversion from graphs to DLGs is reversible.*

**Lemma 2.** *All subtrees in a graph are represented as subtrees in the corresponding DALG. And the conversion from graphs to DALGs is reversible.*

Lemma 1 and Lemmas 2 establish the necessary condition for the effectiveness of iterative message passing on DLGs or DALGs in capturing subtrees of the original graphs. Moreover, these lemmas guarantee that learning from DLGs or DALGs is equivalent to learning from the original graphs.

**Lemma 3.** *DLGN with $L$ layers can express all subtrees of non-isomorphic graphs if the minimum cycle size $S$ and the maximum component radius $R$ of these graphs satisfy $R \leqslant L < S$.*

**Lemma 4.** *ERFGN with $L$ layers can express all subtrees of non-isomorphic graphs if the maximum component radius $R$ of these graphs satisfies $R \leqslant L$.*

The graph radius determines the minimum number of message passing iterations required to transfer information from all nodes in a connected graph to a certain node, and to express all subtrees of the connected graph. Additionally, the effective representation of an unconnected graph requires the ability to express the graph component with the largest radius. Lemma 3 and Lemma 4 provide the necessary conditions to achieve the full potential expressiveness of DLGN and ERFGN, respectively. Moreover, Lemma 4 shows that ERFGN can separate arbitrary non-isomorphic graphs for two reasons: (1) Every graph has at least one path that is not part of any graph that is not isomorphic to that graph. (2) The maximum component radius of a given graph set is finite.

**Lemma 5.** *ERFGN$^{\circ}$ can express all connected subgraph with a cycle.*

**Lemma 6.** *ERFGN$^{\circ}$+GloAttNC can generate identifiable representations for nodes and cycles.*

**Lemma 7.** *Subgraph expressiveness can be ranked as follows: typical MPNNs = DLGN < ERFGN$\simeq$ RFGNN (Chen et al., 2022) < $k$-GNNs using 3-WL (Bodnar et al., 2021a) < ERFGN$^{\circ}$< ERFGN$^{\circ}$+GloAttNC.*

These lemmas show that cycle modeling and GloAttNC progressively enhance the expressiveness of ERFGN. ERFGN and ERFGN$^{\circ}$ fail to achieve identifiable node or cycle representations because nodes or cycles in non-isomorphic graphs may only connect isomorphic subtrees. To address this limitation, ERFGN$^{\circ}$+GloAttNC considers all subtrees in updating, ensuring that the generated representations capture the isomorphism information of source graphs. The isomorphism information also allows ERFGN$^{\circ}$+GloAttNC to express subgraphs consisting of subcycles and subtrees.

**Tree height and cycle size requirements.** ERFGN with $L$ layers only transmits messages between DALG nodes with a maximum distance of $L$. Therefore, when using ERFGN with $L$ layers, only path trees with heights up to $L$ need to be extracted. Additionally, the maximum size of chordless cycles to be extracted should be larger than $L$.

## 2.7 LESS OVER-SQUASHING

To assess the impact of *over-squashing*, Topping et al. (2022) introduced the *Jacobian* of a MPNN-output to measure the influence exerted by changes in the input features of each node on the output representations of each node. This analysis shows that changes in the input features of certain nodes may have only a slight impact on the output representations of specific nodes, leading to information *over-squashed*. Subsequently, (Chen et al., 2022) extended the *Jacobian* of a MPNN-output to quantify the sensitivity of a MPNN in capturing per-path information, and showed that source paths leading to a smaller number of message passing paths are harder to capture than source paths leading to a larger number of message passing paths. They have also showed that the number of message passing paths generated from a source path is equal to the number of random walks generated from its corresponding path in the surrogate graph. Therefore, typical MPNNs often generate fewer message-passing paths for large subgraphs than for small subgraphs, leading to the *over-squashing* issue. RFGNN (Chen et al., 2022) uses only one message passing path for capturing an *epath*, thus addressing the imbalance in the number of message passing paths generated from different source paths and mitigating the *over-squashing* issue.

ERFGN also performs redundancy-free message passing. As a result, ERFGN exhibits a similar sensitivity to subtree capture as RFGNN. Nevertheless, RFGNN (Chen et al., 2022) faces difficulties in modeling cycles because the information about a cycle's epath can be *over-squashed* by the extensive information contained in a path tree. By explicitly modeling cycles, ERFGN° avoids the problem of over-squashing cycle information present in RFGNN.

## 2.8 COMPLEXITY ANALYSIS

Table 1: Message passing complexity analysed based on nodes and edges.

|  | Typical MPNNs | RFGNN with $L$ layers | DLGN | ERFGN with $L$ layers |
|---|---|---|---|---|
| *redundancy* | *yes* | *free* | *reduced* | *free* |
| nodes | $\mathcal{O}\left(\lvert V\rvert\right)$ | $\mathcal{O}\left(\lvert V\rvert!/(\lvert V\rvert-L-1)!\right)$ | $\mathcal{O}\left(\lvert E\rvert\right)$ | $\mathcal{O}\left(\lvert E\rvert+\lvert V^{\circ}\rvert\times d^{L}\right)$ |
| edges | $\mathcal{O}\left(\lvert V\rvert+\lvert E\rvert\right)$ | $\mathcal{O}\left(\lvert V\rvert!/(\lvert V\rvert-L-1)!\right)$ | $\mathcal{O}\left(\lvert E\rvert^{2}/\lvert V\rvert\right)$ | $\mathcal{O}\left(\lvert E\rvert^{2}/\lvert V\rvert+\lvert V^{\circ}\rvert\times d^{L}\right)$ |

The complexity analysis is divided into two parts: graph conversions and message passing. The DLG conversion, which involves constructing nodes and edges, has linear time and memory complexity. The DALG conversion requires cycle detection, path tree extraction and sub-DALG construction. The listing of $N^{\circ}$ chordless cycles can be done in $\mathcal{O}((\lvert E\rvert+\lvert V\rvert)(N^{\circ}+1))$ time (Dias et al., 2013). Let $V^{\circ}$ be the nodes of the cyclic subgraph, and $d^{\circ}$ be the maximum node degree. Then, $\lvert V^{\circ}\rvert$ path trees are extracted, where each node at worst has $d^{\circ}$ child nodes. Therefore, the extracted path trees consist of $(\lvert V^{\circ}\rvert\times d^{L})$ nodes and edges at worst. Thus, for ERFGN with $L$ layers, the path tree extraction, as well as the sub-DALG construction, has a complexity of $\mathcal{O}\left(\lvert E\rvert+\lvert V^{\circ}\rvert\times d^{L}\right)$. The message passing complexity of MPNNs grows linearly with the size of the surrogate structures. Given that the real graphs we are interested in are sparse and the cyclic subgraphs are small, the DALG conversion is efficient, and the constructed DALGs are small. The comparative complexity analysis of different message passing schemes is given in Table 1. This analysis shows that our models offer improved efficiency in addressing message passing redundancy. The cycle modeling complexity is $\mathcal{O}\left(N^{\circ}\right)$, while GloAttNC has a complexity of $\mathcal{O}\left(\lvert V\rvert^{2}\right)$. ERFGN°+GloAttNC is more efficient than graph transformers because GloAttNC performs global attention only once.

## 3 EXPERIMENT

In this section, we conduct experiments on total 11 datasets, including synthetic and real-world tasks, to verify the performance of our models. The statistical details of all datasets can be found in

the Appendix D. Additional experimental results and ablation studies are included in Appendix F. The source code is provided in `https://anonymous.4open.science/r/ERFGN-3863`.

The baseline methods against which our models are compared are as follows: **Popular MPNNs** such as GIN (Xu et al., 2019), GatedGCN (Dwivedi et al., 2023), PNA (Corso et al., 2020). **Avanced GNNs** such as $k$-GNNs (Morris et al., 2019), PPGN (Maron et al., 2019a), CIN (Bodnar et al., 2021a), CRaWL (Toenshoff et al., 2021), GIN-AK+ (Zhao et al., 2022), NGNN (Zhang & Li, 2021), KP-GIN′ (Feng et al., 2022), I$^2$GNN (Huang et al., 2022), PathNNs Michel et al. (2023). And **graph transformers** such as SAN (Kreuzer et al., 2021), Graphormer (Ying et al., 2021), EGT (Hussain et al., 2022), GPS (Rampášek et al., 2022).

Table 2: Test performance on MNIST and CIFAR10. **Bold** ones are the best.

| Model | MNIST | CIFAR10 |
|---|---|---|
| | Accuracy ($\uparrow$) | Accuracy ($\uparrow$) |
| GIN | $96.485 \pm 0.252$ | $55.255 \pm 1.527$ |
| GatedGCN | $97.340 \pm 0.143$ | $67.312 \pm 0.311$ |
| PNA | $97.94 \pm 0.12$ | $70.35 \pm 0.63$ |
| CRaWl | $97.944 \pm 0.050$ | $69.013 \pm 0.259$ |
| GIN-AK+ | - | $72.19 \pm 0.13$ |
| EGT | $98.173 \pm 0.087$ | $68.702 \pm 0.409$ |
| GPS | $98.051 \pm 0.126$ | $72.298 \pm 0.356$ |
| DLGN | $\mathbf{98.640 \pm 0.052}$ | $\mathbf{73.386 \pm 0.312}$ |

Table 3: Test performance on Peptides-func and Peptides-struct. **Bold** ones are the best. Underlined ones are better than baselines.

| Model | Peptides-func (AP ($\uparrow$)) | Peptides-struct (MAE$\downarrow$) |
|---|---|---|
| GINE | $0.5498 \pm 0.0079$ | $0.3547 \pm 0.0045$ |
| GatedGCN | $0.5864 \pm 0.0077$ | $0.3420 \pm 0.0013$ |
| PathNNs | $0.6816 \pm 0.0026$ | $0.2540 \pm 0.0046$ |
| Transformer+LapPE | $0.6326 \pm 0.0126$ | $0.2529 \pm 0.0016$ |
| SAN+LapPE | $0.6384 \pm 0.0121$ | $0.2683 \pm 0.0043$ |
| SAN+RWSE | $0.6439 \pm 0.0075$ | $0.2545 \pm 0.0012$ |
| GPS | $0.6535 \pm 0.0041$ | $0.2500 \pm 0.0005$ |
| DLGN | $\underline{0.6764 \pm 0.0055}$ | $0.2540 \pm 0.0008$ |
| ERFGN | $\underline{0.6790 \pm 0.0055}$ | $0.2553 \pm 0.0028$ |
| ERFGN$^\circ$ | $\underline{0.6869 \pm 0.0056}$ | $0.2563 \pm 0.0011$ |
| ERFGN$^\circ$+GloAttNC | $\underline{\mathbf{0.6912 \pm 0.0049}}$ | $\underline{\mathbf{0.2468 \pm 0.0014}}$ |

**MNIST** and **CIFAR10** (Dwivedi et al., 2023) contain large-size graphs converted from images. Due to the large and dense nature of these graphs, we only apply a 3-layer DLGN to these datasets. The experimental setup follows that of GPS (Rampášek et al., 2022): $\sim 100K$ parameter budget. The results are shown in Table 2; the mean $\pm$ s.d. of the results from 5 runs is reported. Although not exceeding the expressiveness of typical MPNNs or higher-order MPNNs, DLGN benefits from the mitigation of *over-squashing*, which helps DLGN to achieve the best performance on both datasets. In addition, DLGN shows high efficiency on both datasets: The pre-computation times for converting the source graphs of MNIST and CIFAR10 into DLGs are 41.43s and 65.15s, respectively. And the per-epoch/total training time is 40.0s/1.1h – showing that our models are efficient.

**Peptides-func** and **Peptides-struct** datasets from Long-Range Graph Benchmarks (Dwivedi et al., 2022) are specifically designed to test the capability of models in performing tasks that require reasoning with long-range interactions. The the experimental setup follows that of GPS (Rampášek et al., 2022): $\sim 500K$ parameter budget. Results are shown in Table 3; Shown is the mean $\pm$ s.d. of 5 runs. Our methods obtain the best performance on both datasets, outperforming SOAT graph transformers – showing that our models are capable of learning long range interactions. In addition, the pre-computation process of constructing surrogate graphs can be finished in 40s, and the per-epoch/total training time of the specific method achieved the best result is 10s/10m. – showing that our models are efficient for both tasks.

Table 4: MAE results on QM9 (smaller the better). **Bold** ones are the best.

| Target | $\mu$ | $\alpha$ | $\epsilon_{HOMO}$ | $\epsilon_{LUMO}$ | $\Delta_\epsilon$ | $\langle R^2 \rangle$ | ZPVE | $U_0$ | $U$ | $H$ | $G$ | $C_v$ |
|---|---|---|---|---|---|---|---|---|---|---|---|---|
| DTNN | 0.224 | 0.95 | 0.00388 | 0.00512 | 0.0112 | 17 | 0.00172 | 2.43 | 2.43 | 2.43 | 2.43 | 0.27 |
| PPGN | 0.231 | 0.382 | 0.00276 | 0.00287 | 0.00406 | 16.7 | 0.00064 | 0.234 | 0.234 | 0.229 | 0.238 | 0.184 |
| NGNN | 0.433 | 0.265 | 0.00279 | 0.00276 | 0.0039 | 20.1 | 0.00015 | 0.205 | 0.2 | 0.249 | 0.253 | 0.0811 |
| KP-GIN′ | 0.358 | 0.233 | 0.0024 | 0.00236 | 0.00333 | 16.51 | 0.00017 | **0.0682** | **0.0696** | **0.0641** | **0.0484** | 0.0869 |
| I$^2$GNN | 0.428 | 0.23 | 0.00261 | 0.00267 | 0.0038 | 18.64 | **0.00014** | 0.211 | 0.206 | 0.269 | 0.261 | **0.073** |
| ERFGN$^\circ$+ GloAttNC | **0.1889** | **0.198** | **0.00218** | **0.00207** | **0.00316** | **5.366** | 0.00016 | 0.113 | 0.14 | 0.17 | 0.141 | 0.076 |

**QM9** (Ramakrishnan et al., 2014; Wu et al., 2018) contains over 130K molecules with 12 different regression task. The dataset is split into training, validation, and test sets with a ratio of $0.8/0.1/0.1$.

Table 5: Test performance on ZINC Subset and OGBG-MOLHIV. **Bold** ones are the best.

| Model | ZINC Subset MAE↓ | OGBG-MOLHIV AUROC (↑) |
|---|---|---|
| GIN | $0.526 \pm 0.051$ | - |
| GIN+virtual node | - | $0.7707 \pm 0.0149$ |
| GatedGCN | $0.282 \pm 0.015$ | - |
| GSN | $0.101 \pm 0.010$ | - |
| GSN (directional) | - | $0.8039 \pm 0.0090$ |
| GSN(GIN+VN base) | - | $0.7799 \pm 0.0100$ |
| CIN | $0.079 \pm 0.006$ | $\mathbf{0.8094 \pm 0.0057}$ |
| CRaW1 | $0.085 \pm 0.004$ | - |
| GIN-AK+ | $0.080 \pm 0.001$ | $0.7961 \pm 0.0119$ |
| PathNNs | $0.090 \pm 0.004$ | $0.7917 \pm 0.0109$ |
| SAN | $0.139 \pm 0.006$ | $0.7785 \pm 0.2470$ |
| Graphormer | $0.122 \pm 0.006$ | - |
| EGT | $0.108 \pm 0.009$ | - |
| Graphormer-GD | $0.081 \pm 0.009$ | - |
| GPS | $0.070 \pm 0.004$ | $0.7880 \pm 0.0101$ |
| ERFGN°+ GloAttNC | $\mathbf{0.068 \pm 0.002}$ | $0.7898 \pm 0.0166$ |

Table 6: Accuracy on TUDatasets. **Bold** ones are the best. Underlined ones are better than baselines. (Higher is better.)

| | ENZYMES | PROTEINS_full | FRANKENSTEIN | NCI1 |
|---|---|---|---|---|
| GIN | $72.5 \pm 6.1$ | $76.2 \pm 2.8$ | $64.6 \pm 2.9$ | $82.7 \pm 1.7$ |
| $k$-GNN | $54.7 \pm 6.8$ | $77.3 \pm 4.2$ | $68.3 \pm 1.8$ | $61.0 \pm 5.3$ |
| PPGN | $63.3 \pm 11.8$ | $77.2 \pm 4.7$ | $64.6 \pm 2.9$ | $83.2 \pm 1.1$ |
| CIN | $62.6 \pm 8.1$ | $77.0 \pm 4.3$ | $74.4 \pm 1.3$ | $83.6 \pm 1.4$ |
| PathNNs | $73.0 \pm 5.2$ | $75.2 \pm 3.9$ | $73.4 \pm 1.5$ | $82.3 \pm 1.9$ |
| RFGNN | $\mathbf{75.3 \pm 5.0}$ | $77.5 \pm 3.4$ | $74.5 \pm 1.3$ | $83.6 \pm 1.6$ |
| ERFGN° | $73.5 \pm 5.9$ | $\underline{77.5 \pm 2.0}$ | $\underline{77.2 \pm 1.7}$ | $\underline{\mathbf{86.4 \pm 1.7}}$ |
| ERFGN°+ GloAttNC | $74.8 \pm 5.1$ | $\underline{\mathbf{78.4 \pm 3.9}}$ | $\underline{\mathbf{78.4 \pm 2.0}}$ | $\underline{\mathbf{86.6 \pm 1.7}}$ |

**TUDataset** (Morris et al., 2020a) contain small or medium molecular graph datasets. The setting of the experiment follows that of (Xu et al., 2019). The accuracy results are listed in Table 6; the mean $\pm$ s.d. of results from 10 runs is shown. The experimental results show the high performance of our models.

We report the single-time test MAE in Table 4, which shows our model, ERFGN°+GloAttNC using the '1-N' scheme, outperforms all baseline methods in several targets.

**ZINC Subset** and **OGBG-MOLHIV** are both molecular graphs. The training setup and the evaluation procedure follow those of (Bodnar et al., 2021a). On the ZINC Subset, ERFGN°+GloAttNC outperforms the baselines, while on OGBG-MOLHIV ERFGN° achieves close to the best results.

## 4    CONCLUSION

In this study, we investigate the nature of the *message passing redundancy* problem and present efficient solutions to the problem. First, we propose DLGN, which uses the DLG surrogate structure to reduce redundancy caused by self-loops and backtrackings. Second, we propose ERFGN, which uses the DALG surrogate structure to perfectly remove all redundancy. By removing all redundancy, ERFGN addresses the *message confusion* problem and expresses arbitrary subtrees. To achieve higher expressiveness and less over-squashing, we propose ERFGN°, which provides improved expressiveness of subgraphs with cycles by extending ERFGN with a module that explicitly models cycles. And we propose GloAttNC to model **Glo**bal **Att**tion of **N**odes and **C**ycles, which allows ERFGN°+GloAttNC to generate identifiable representations for nodes and cycles, and to express a wider range of subgraphs. Theoretical analysis shows that our ERFGN° and ERFGN°+GloAttNC models are more expressive than higher-order GNNs using the 3-WL test. And empirical results support the high efficiency of our models on multiple graph datasets. Ablation studies on several graph datasets confirm the theoretically improved expressiveness of our models.

Despite the success of ERFGN°+GloAttNC on realistic datasets, there still be room for improvement in the expressiveness due to the fact that ERFGN°+GloAttNC ignores connectivity between subtrees/subgraphs, such as distance, number of connections, etc., when modeling global attention. In addition, some adaptations of our methods are needed in order to apply them to graphs with dense connections or a high number of subrings. One possible solution could be to decompose the graph, as ESAN (Bevilacqua et al., 2022) does, into several subtrees, which will be investigated in our future work.

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

# A  RELATED WORK

Xu et al. (2019) and Morris et al. (2019) investigated the theoretical analysis on expressive power of GNNs and claimed 1-Weisfeiler-Lehman (1-WL) test (Leman & Weisfeiler, 1968) as an upper bound of typical aggregation-based GNNs in distinguishing graph isomorphism. To develop GNNs with higher expressiveness, extensive methods have been proposed, which can be broadly classified into the following categories:

**$k$-WL Hierarchy.** One straightforward strategy to design provably more powerful GNNs is to take inspiration from the higher-order WL tests. Instead of updating node representations, these higher-order GNNs compute a representation for each $k$-tuple of nodes ($k \geqslant 2$) by aggregating features from different node tuples (Morris et al., 2019; Maron et al., 2019a;b;c; Keriven & Peyré, 2019; Azizian & Lelarge, 2021; Geerts & Reutter, 2021). To overcome the huge computational cost of higher-order message passing, several recent works considered improving efficiency by leveraging the sparse and local nature of graphs (Morris et al., 2020b; 2022).

**Substructure-based GNNs.** Another way to design more expressive GNNs is to take inspiration from the failed cases of 1-WL test. In particular, Chen et al. (2020) pointed out that standard MPNNs cannot detect/count common substructures such as cycles, cliques, and paths. Following this finding, Bouritsas et al. (2020) designed the Graph Substructure Network (GSN) by injecting substructure counting into node features. Further, Bodnar et al. (2021b;a); Thiede et al. (2021); Horn et al. (2021) developed novel WL update schemes that take into account these substructures (e.g., cycles or cliques). Particularly, MPSNs (Bodnar et al., 2021b) and CWNs (Bodnar et al., 2021a) have been shown to be no less powerful than the 3-WL test.

**Subgraph-based GNNs** It has been observed that graphs that are indistinguishable by the 1-WL algorithm often exhibit a significant degree of symmetry (Kreuzer et al., 2021). Building upon this observation, recent approaches have aimed to break this symmetry by feeding subgraphs into the message passing process. To ensure equivariance, these subgraphs are generated symmetrically from the original graph using predefined policies, and the final output is aggregated across all subgraphs. Previous works have proposed various subgraph generation policies, including node deletion (Cotta et al., 2021), edge deletion (Bevilacqua et al., 2022), and ego-networks (Zhao et al., 2022; Zhang & Li, 2021; You et al., 2021). Bevilacqua et al. (2022); Frasca et al. (2022) developed unified frameworks which includes per-layer aggregation across subgraphs.

**Graph Transformers** Graph Transformers, which have gained popularity recently, are also solutions to over-smoothing (Oono & Suzuki, 2020) and over-squashing (Alon & Yahav, 2021; Topping et al., 2022; Di Giovanni et al., 2023). Graph Transformers allow nodes to attend to all other nodes in a graph by using global attention, but this requires nodes to be better identifiable within the graph and its substructures (Dwivedi & Bresson, 2021). Dwivedi & Bresson (2021) proposed a Transformer architecture involving a global attention for neighborhood interaction of each node with Laplacian Positional Encoding (LapPE). Kreuzer et al. (2021) proposed a learned positional encoding (LPE) and added the position information of each node to fully-connected Transformer that did not undergo the over-squashing. Rampášek et al. (2022) summarized the different types of positional or structural encodings. However, they found there is no one-size-fits-all solution for all datasets.

Reducing message redundancy has been investigated in several previous studies. Both Mahé et al. (2004) and Chen et al. (2018) adopted directed line graphs to avoid backtracking. Specifically, Mahé et al. (2004) proposed a kernel function that transformed the input graphs into directed ones and generated a new graph with node set consisting of original vertices and line nodes connect with directed edges. The transformed graphs well captured the structural information and improved the ability to graph classification. However, the generated node set combined with vertices and edges would increase computational cost. And Chen et al. (2018) proposed to address node representation learning on community detection by an efficient neural network that transformed the input graph to the line graph avoiding back-trackings. The Line Graph Neural Networks (LGNN) iteratively aggregated to learn the high-order interactions among nodes in the graph. Their idea is similar with our DLGN to address the redundant message passing issue. However, LGNN and DLGN are not sufficient to well address the native cycles and achieve superior performance on our tasks. RFGNN (Chen et al., 2022) is a pioneering study that achieves redundancy-free message passing. RFGNN conducts message passing on a surrogate structure called TPTs, which consist of extended paths (epaths) extracted from source graphs, where an epath can represent a cycle. The use of TPTs ensures that only one mes-

sage passing path captures an epath, making RFGNN redundancy free. Yang et al. (2019) proposed a path-based node representation learning (SPAGAN) as a generalized version of GAT to address node classification tasks. The node representation learned the global interaction by aggregating shortest paths to high-order neighbors under path length controlling. Alsentzer et al. (2020) proposed three kinds of paths, i.e., Shortest Path(SP), all Shortest Path($SP^+$), and A Simple Path(AP) to build path trees for each node. However, the extraction of path trees suffer from high computational cost.

## B    DETAILS OF THE GRAPH CONVERSIONS

### B.1    DLG CONVERSION ALGORITHM

---

**Algorithm 1:** DLG_Conversion

**Input:** a graph $\mathcal{G}=(V, E)$

1  $DLG\_V \leftarrow Arange(|E|)$      ▷ nodes of target DLG
2  $D \leftarrow \text{Dest}(E)$      ▷ destination nodes of $\mathcal{G}$'s edges
3  $S \leftarrow \text{Source}(E)$      ▷ source nodes of $\mathcal{G}$'s edges
4  $NE \leftarrow \text{SparseMatrix}(row=S, col=Arange(|E|))$      ▷ node-edge incidence matrix
5  $EN \leftarrow \text{SparseMatrix}(col=S, row=Arange(|E|))$      ▷ edge-node incidence matrix
6  $EE \leftarrow \text{SparseMatMul}(EN, NE)$      ▷ edge-edge adjacency matrix
7  $EE\_dst \leftarrow \text{Dest}(EE)$      ▷ destination edges of edge-edge connections
8  $EE\_src \leftarrow \text{Source}(EE)$      ▷ source edges of edge-edge connections
9  $EE\_D \leftarrow D[EE\_dst]$      ▷ destination nodes of destination of edge-edge connections
10  $EE\_S \leftarrow D[EE\_src]$      ▷ source nodes of source of edge-edge connections
11  $Mask \leftarrow EE\_D \neq EE\_S$      ▷ mask of non-backtracking edge-edge connections
12  $DLG\_D \leftarrow EE\_D[Mask]$      ▷ destination nodes of target DLG's edges
13  $DLG\_S \leftarrow EE\_S[Mask]$      ▷ source nodes of target DLG's edges
14  $DLG\_E \leftarrow \text{CreateAdjacency}(DLG\_D, DLG\_S)$      ▷ adjacency matrix of target DLG
15  $DLG \leftarrow \text{CreateGraph}(DLG\_V, DLG\_E)$      ▷ target DLG
16  **for** *each node $v$ in $V$* **do**
17      Create a set $\mathbf{D}(v) = \{\mathbf{u} \mid EE\_dst[\mathbf{u}] == v, \forall \mathbf{u}\}$

**Output:** $DLG$

---

The DLG conversion method constructs a DLG from an original graph. This is achieved by transforming each directed edge of the original graph into a DLG node, and creating a DLG edge matrix to represent the connections between pairs of nodes. The process is described in Algorithm 1, which utilizes sparse matrix operations. Specifically, the SparseMatrix operation constructs a sparse matrix, while the SparseMatMul operation performs sparse matrix multiplication. Both of these operations are supported by the PyTorch Sparse Library [4]. Additionally, the Arange($stop$) operation generates a vector with values ranging from 0 to $stop - 1$. SparseMatMul($EN, NE$) is the most time-consuming step in this algorithm, whose complexity is linear in the number of non-zero elements in sparse matrices $EN$ and $NE$ (Yuster & Zwick, 2005), i.e., $\mathcal{O}(|E|)$. The most memory-consuming data utilized in this algorithm is the edge size of the DLG, whose memory complexity is $\mathcal{O}(|E|^2/|V|)$. Consequently, the DLG conversion algorithm is efficient.

### B.2    DALG CONVERSION ALGORITHM

**Definition 2** (Path Tree). *Given a graph $\mathcal{G} = (V, E)$ and a node $u \in V$, the height-L path tree $\mathcal{T}$ is constructed by searching all paths with length up to L, ending at node $u$. In the path tree $\mathcal{T}$, the root node is $u$, and a node is connected to its parent node if and only if there is an edge connecting them in the original graph, and this node is not repeated in the path to the root node.*

Our DALG conversion algorithm incorporates the path tree structure, which is designed to capture all paths ending at a node, allowing the expression of all subtrees rooted at that node. To introduce the DALG conversion algorithm, we first give the formal definition of the path tree structure in Definition 2. And then we describe the algorithm for extracting path trees in Algorithm 2. Chen et al.

---

[4]https://github.com/rusty1s/pytorch_sparse

---

**Algorithm 2:** ExtractPathTrees (Notice that two different tree nodes are congruent ($\cong$) if and only if they are derived from the same original node.)

---

**Input:** a graph $\mathcal{G} = (V, E)$, the tree height $L$

1  **Declare** $\mathcal{T}s$ is a list
2  **for** $v$ *in* $V$ **do**
3     **Declare** $\mathcal{T}$ is a tree composed of the node $v$
4     **while** $\mathcal{T}$ *'s height* $< L$ **do**
5         **Declare** $U$ is the set of nodes at $\mathcal{T}$'s bottom level
6         **for** $u$ *in* $U$ **do**
7             **Declare** $p$ be the path from $v$ to $u$ in $\mathcal{T}$
8             **for** $q$ *in neighbors of* $u$ *in* $\mathcal{G}$ **do**
9                 **if** *p consists of no node* $\cong q$ **then**
10                    Attach $q$ to $\mathcal{T}$ with $u$ as its parent

11     Append $T$ to $\mathcal{T}s$

**Output:** $\mathcal{T}s$

---

**Algorithm 3:** DALG_Conversion

---

**Input:** a graph $\mathcal{G} = (V, E)$, $size\_bound$

1  $C \leftarrow$ FindChordlessCycles($E, size\_bound$)
2  **if** $len(C) == 0$ **then**
3     $DALG \leftarrow$ DLG_Conversion($\mathcal{G}$)                          $\triangleright$ no cycles are found
4  **else**
5     $CSg \leftarrow (V(cycles), E(cycles))$          $\triangleright$ cyclic subgraph consisting of all cycles
6     $ASg \leftarrow (V(E - E(cycles)), E - E(cycles))$     $\triangleright$ complementary acyclic subgraph
7     $J \leftarrow (V(CSg) \cap V(ASg))$                 $\triangleright$ joint nodes of $CSg$ and $ASg$
8     $\mathcal{D}^\tau \leftarrow$ DLG_Conversion($ASg$)           $\triangleright$ sub-DALG extracted from $ASg$
9     $\mathcal{T}s \leftarrow$ ExtractPathTrees($CSg, size\_bound$)        $\triangleright$ path trees extracted from $CSg$
10     $\mathcal{E} \leftarrow$ RootEdges($\mathcal{T}s$)       $\triangleright$ DALG nodes generated from root edges of path trees
11     $\mathcal{D}^\circ \leftarrow$ Tree2DALG_Conversion($\mathcal{T}s$)      $\triangleright$ sub-DALG extracted from $CSg$
12     $DALG \leftarrow$ Connect($\mathcal{D}^\tau, \mathcal{D}^\circ$)           $\triangleright$ connection of $\mathcal{D}^\tau$ and $\mathcal{D}^\circ$
13     **for** $v$ *in* $V$ **do**
14         Create a set $\mathbf{D}(u) = \{v \to u \mid (v, u) \in \mathcal{G}^\tau \cup (v, u) \in \mathcal{E}\}$.
15     **for** $c$ *in* $C$ **do**
16         Create a set $\mathbf{C}(c) = \{v \to u \mid (v, u) \in c \cap (v, u) \in \mathcal{E}\}$.

**Output:** $DALG$

---

(2022) adopted a similar path tree structure for message passing, i.e., the TPT. However, it is important to highlight that the path tree structure used in this study is different from the TPT structure. The TPT structure permits the two endpoints of a path within the trees to be identical (extracted from the same original node), thus enabling the representation of a cycle using a path. However, in this study, instead of representing a cycle by a path, we explicitly express cycle-based subgraph using our proposed cycle modeling module. The DALG conversion is described in Algorithm 3. In this algorithm, the function FindChordlessCycles($E, size\_bound$) [5] lists cycles with size up to $size\_bound$ in the given graph. And, the function Tree2DALG_conversion($\mathcal{T}s$) constructs a sub-DALG from the given $\mathcal{T}s$ by following a two-step process: First, it converts each directed edge of $\mathcal{T}s$ into a DALG node. Then, it adds edges between every pair of DALG nodes that correspond to connected edges in the original $\mathcal{T}s$. Additionally, the function Connect($\mathcal{D}^\tau, \mathcal{D}^\circ$) connects $\mathcal{D}^\tau$ and $\mathcal{D}^\circ$ to form the desired DALG through a process that has been introduced in step (f) in Section 2.3. The DALG conversion algorithm is efficient for the following reasons: (1) The listing of $N^\circ$ chordless cycles can be done in $\mathcal{O}((|E| + |V|)(N^\circ + 1))$ time (Dias et al., 2013). (2) The time complexity and memory complexity of constructing sub-DALG $\mathcal{D}^\tau$ are $\mathcal{O}(|E|)$ and $\mathcal{O}(|E|^2/|V|)$, respectively.

---

[5] Supported by the NetworkX library

(3) The time complexity and memory complexity of the construction of sub-DALG $\mathcal{D}^\circ$ are both $\mathcal{O}\left(|E| + |V^\circ| \times d^\circ!\right)$, where $V^\circ$ be the nodes of the cyclic subgraph, and $d^\circ$ be the maximum node degree. (4) Other steps is done in linear complexity.

## C  EXPRESSIVE POWER ANALYSIS

This section provides a comprehensive analysis of the theoretical expressiveness of our models, namely DLGN, ERFGN, ERFGN$^\circ$ and ERFGN$^\circ$+GloAttNC. DLGN and ERFGN employ a two-step process for generating representations of nodes and graphs. Initially, the input graphs undergo a transformation into surrogate structures, specifically DLGs or DALGs. Subsequently, message passing procedure is applied to these surrogate structures to produce node and graph representations. Furthermore, ERFGN$^\circ$ adds a cycle modeling module to ERFGN. And ERFGN$^\circ$+GloAttNC augments ERFGN$^\circ$ with GloAttNC.

To establish the foundation for our analysis of expressiveness, we first provide a restatement of the concept of 'graph isomorphism'. Following it, we show that all subtrees in a graph are represented as subtrees in the corresponding surrogate graph. This stablish the necessary condition for the effectiveness of iterative message passing on DLGs or DALGs in capturing subtrees of the source graphs. Moreover, we prove the invertibility of the conversion from graphs to DLGs or DALGs, ensuring that learning from DLGs or DALGs is equivalent to learning from the original graphs.

We proceed to establish the necessary conditions under which our models can generate identifiable representations for non-isomorphic subtrees, as well as for non-isomorphic graphs. The first essential condition is the injectiveness of our models, which guarantees that distinct inputs are mapped to distinguishable outputs. The second crucial condition is the sufficient layers of our models, enabling them to capture all subtrees effectively.

It is important to note that injectiveness alone may not be sufficient to differentiate all potential non-isomorphic subtrees within surrogate structures. This limitation arises due to the message confusion issue and the constrained receptive field of our models. In certain scenarios, our models may encounter identical inputs from non-isomorphic surrogate structures, which hinders their ability to discriminate between them effectively. To illustrate the impact of the limited receptive field, let us consider a scenario where models consist of only one layer. In such cases, the receptive field is confined solely to the node initial representations, resulting in the inability of models to distinguish non-isomorphic surrogate structures that share the same set of node initial representations.

Therefore, it becomes crucial to analyze the conditions under which our models can possess a sufficient receptive field. By satisfying all the aforementioned conditions, our models have the potential to accurately differentiate all possible non-isomorphic surrogate structures and, consequently, non-isomorphic original graphs.

### C.1  DEFINITION OF GRAPH ISOMORPHISM

**Definition 3** (Graph isomorphism). *Two graphs $\mathcal{G}_1$ and $\mathcal{G}_2$ are isomorphic, if and only if there exists at least one bijection (one-to-one correspondence) between their node sets, denoted as $\mathcal{I} : V(\mathcal{G}_1) \to V(\mathcal{G}_2)$. The bijection guarantees that any two nodes $u$ and $v$ in $\mathcal{G}_1$ are adjacent if and only if their corresponding nodes $\mathcal{I}(u)$ and $\mathcal{I}(v)$ are adjacent in $\mathcal{G}_2$. Moreover, for labeled graphs, this bijection ensures that the labels of node $u$ and node $\mathcal{I}(u)$ are equal, and the labels of edge $(u, v)$ and edge $(\mathcal{I}(u), \mathcal{I}(v))$ are equal too.*

### C.2  LEARNING FROM DLGS OR DALGS EQUALS LEARNING FROM ORIGINAL GRAPHS

The accurate prediction of graph properties heavily relies on the ability of GNNs to effectively capture a wide range of non-isomorphic subgraphs and generate distinguishable representations for them. To accomplish this, the surrogate structures, the DLG and the DALG, are meticulously designed with specific purposes in mind. These objectives include ensuring that all subtrees present in a graph are accurately represented as subtrees in the corresponding DLG or DALG, so that iterative message passing on DLGs or DALGs can capture all subtrees within the original graphs. Additionally, it is crucial that the conversion process from graphs to DLGs or DALGs is reversible, meaning that the modeling of the original graph can be precisely replaced by the modeling of its correspond-

ing DLG or DALG. The fulfillment of these goals is guaranteed by Lemma 1 and Lemma 2, which are proven below.

**Lemma 1.** *All subtrees in a graph are represented as subtrees in the corresponding DLG. And the conversion from graphs to DLGs is reversible.*

*Proof.* We first prove that all subtrees in a graph are represented as subtrees in the corresponding DLG. Because a directed edge in an original graph only generates a DLG node, for each path $a \to b \to c \to \cdots \to y \to z$ in an original graph, there exists one corresponding path $\mathbf{a} \to \mathbf{b} \to \cdots \to \mathbf{z}$ in the corresponding DLG, where $\mathbf{a}$ is generated from $a \to b$, $\mathbf{b}$ is generated from $b \to c$, ..., and $\mathbf{z}$ is generated from $y \to z$. Therefore, all paths in a graph are represented as paths in the corresponding DLG. A tree can be represented as organized paths. Then, each subtree in a graph is represented as a subtree which consists of DLG paths and a root node to which all these DLG paths virtually connect.

Moving forward, we prove that the conversion from graphs to DLGs is reversible. The invertibility of this conversion is guaranteed with that two DLGs are isomorphic if and only if their corresponding original graphs are isomorphic. Let $\mathcal{G}_1$ and $\mathcal{G}_2$ be two graphs, and $\mathcal{D}_1$ and $\mathcal{D}_2$ be their corresponding DLGs.

If $\mathcal{G}_1$ and $\mathcal{G}_2$ are isomorphic, there exists at least one bijection $\mathcal{I} : V(\mathcal{G}_1) \to V(\mathcal{G}_2)$. This bijection guarantees that any path $(a, b, c)$ in $\mathcal{G}_1$ corresponds to a path $(\mathcal{I}(a), \mathcal{I}(b), \mathcal{I}(c))$ in $\mathcal{G}_2$, where $a \neq c$ and $\mathcal{I}(a) \neq \mathcal{I}(c)$. Therefore, there exists a bijection between the node sets of $\mathcal{D}_1$ and $\mathcal{D}_2$, denoted as $\mathcal{I}^\iota : V(\mathcal{D}_1) \to V(\mathcal{D}_2)$. This bijection ensures that each DLG node $\mathbf{u}$ generated from a directed edge $(a, b)$ in $\mathcal{G}_1$ corresponds to a DLG node $\mathcal{I}^\iota(\mathbf{u})$ generated from the directed edge $(\mathcal{I}(a), \mathcal{I}(b))$ in $\mathcal{G}_2$. This bijection also guarantees that any two nodes $\mathbf{u}$ and $\mathbf{v}$ in $\mathcal{D}_1$ are connected if and only if their corresponding nodes $\mathcal{I}^\iota(\mathbf{u})$ and $\mathcal{I}^\iota(\mathbf{v})$ are connected in $\mathcal{D}_2$. Consequently, $\mathcal{D}_1$ and $\mathcal{D}_2$ are isomorphic.

Conversely, if $\mathcal{D}_1$ and $\mathcal{D}_2$ are isomorphic, there exists a bijection $\mathcal{I}^\iota : V(\mathcal{D}_1) \to V(\mathcal{D}_2)$, which ensures that each DLG node $\mathbf{u}$ in $\mathcal{D}_1$ corresponds to a DLG node $\mathcal{I}^\iota(\mathbf{u})$ in $\mathcal{D}_2$. This bijection also guarantees that any two nodes $\mathbf{u}$ and $\mathbf{v}$ in $\mathcal{D}_1$ are connected if and only if their corresponding nodes $\mathcal{I}^\iota(\mathbf{u})$ and $\mathcal{I}^\iota(\mathbf{v})$ are connected in $\mathcal{D}_2$. Adjacent DLG nodes must be generated from adjacent source edges, because each source directed edge is converted into a unique DLG node. Consequently, there exists bijection between the edge sets of $\mathcal{G}_1$ and $\mathcal{G}_2$. This bijection ensures that any two edges in $\mathcal{G}_1$ are connected if and only if the corresponding edges in $\mathcal{G}_2$ are connected. Hence, this bijection ensures that any two nodes in $\mathcal{G}_1$ are connected if and only if the corresponding nodes in $\mathcal{G}_2$ are connected. Consequently, $\mathcal{G}_1$ and $\mathcal{G}_2$ are isomorphic.

Consequently, two DLGs are isomorphic if and only if their corresponding original graphs are isomorphic. Thus, the conversion from graphs to DLGs is reversible. □

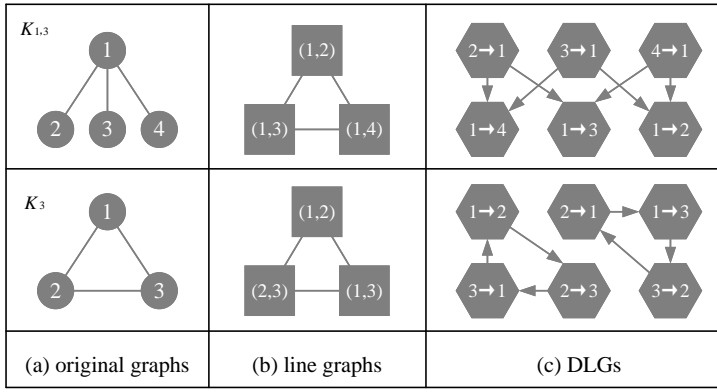

Figure 4: DLGs of $K_3$ and $K_{1,3}$ are non-isomorphic.

By the way, DLGs are a more powerful surrogate than line graphs. This is because the conversion of graphs into line graphs is not always reversible. According to the Whitney graph isomorphism

theorem (Whitney, 1992), two connected graphs are isomorphic if and only if their line graphs are isomorphic, except for a single instance: $K_3$, which refers to the complete graph on three vertices, and the complete bipartite graph (a.k.a the claw graph) $K_{1,3}$. These two graphs are not isomorphic, yet they both have $K_3$ as their line graph. However, the DLGs of $K_3$ and $K_{1,3}$ are non-isomorphic, as illustrated in Fig. 4.

**Lemma 2.** *All subtrees in a graph are represented as subtrees in the corresponding DALG. And the conversion from graphs to DALGs is reversible.*

*Proof.* We first demonstrate that every subtree in a graph is represented as a subtree in its corresponding DALG. The proof is analogous to the proof offered in Lemma 1, where we showed that every subtree in a graph is represented as a subtree in its corresponding DLG.

To establish the invertibility of the conversion from graphs to DALGs, we must show that two DALGs are isomorphic if and only if their corresponding original graphs are isomorphic. Let $\mathcal{G}_1$ and $\mathcal{G}_2$ be two graphs, and $\mathcal{D}_1$ and $\mathcal{D}_2$ be their corresponding DALGs. Additionally, let $\mathcal{G}_i^\circ$ and $\mathcal{G}_i^\tau$ be the cyclic subgraph and acyclic subgraph of graph $\mathcal{G}_i$, respectively, and let $\mathcal{D}_i^\circ$ and $\mathcal{D}_i^\tau$ be the sub-DALGs extracted from subgraph $\mathcal{G}_i^\circ$ and $\mathcal{G}_i^\tau$, respectively, where $i = 1, 2$.

If $\mathcal{G}_1$ and $\mathcal{G}_2$ are isomorphic, then their cyclic subgraphs $\mathcal{G}_1^\circ$ and $\mathcal{G}_2^\circ$ are isomorphic, and their acyclic subgraphs $\mathcal{G}_1^\tau$ and $\mathcal{G}_2^\tau$ are isomorphic too. Then, $\mathcal{D}_1^\tau$ and $\mathcal{D}_2^\tau$ are isomorphic due to Lemma 1. Moreover, $\mathcal{D}_1^\circ$ and $\mathcal{D}_2^\circ$ are isomorphic since isomorphic graphs always yield isomorphic path trees. Additionally, $\mathcal{G}_1^\circ$ and $\mathcal{G}_1^\tau$ may share joint nodes. If this is the case, these shared nodes are bijectively mapped to the joint nodes of $\mathcal{G}_2^\circ$ and $\mathcal{G}_2^\tau$, and the edges between $\mathcal{D}_1^\circ$ and $\mathcal{D}_1^\tau$ are bijectively mapped to the edges between $\mathcal{D}_2^\circ$ and $\mathcal{D}_2^\tau$. A DALG is constructed by connecting the sub-DALG extracted from the cyclic subgraph and the acyclic subgraph via their joint nodes. Since any part of $\mathcal{D}_1$ can be bijectively mapped onto a corresponding part of $\mathcal{D}_2$, there is a bijection between the node sets of $\mathcal{D}_1$ and $\mathcal{D}_2$, denoted as $\mathcal{I}^\iota : V(\mathcal{D}_1) \rightarrow V(\mathcal{D}_2)$. This bijection ensures that any two nodes $\mathbf{u}$ and $\mathbf{v}$ in $\mathcal{D}_1$ are adjacent if and only if their corresponding nodes $\mathcal{I}^\iota(\mathbf{u})$ and $\mathcal{I}^\iota(\mathbf{v})$ are adjacent in $\mathcal{D}_2$. Therefore, $\mathcal{D}_1$ and $\mathcal{D}_2$ are isomorphic.

Conversely, if $\mathcal{D}_1$ and $\mathcal{D}_2$ are isomorphic, we need to show that $\mathcal{G}_1$ and $\mathcal{G}_2$ are also isomorphic. Firstly, we prove that the DALG $\mathcal{D}_i$ can be decomposed into two sub-DALGs, $\mathcal{D}_i^\tau$ and $\mathcal{G}_i^\circ$. For each node $u \in \mathcal{G}_i$, there exists a set $\mathbf{D}(u)$ containing all DALG nodes that point to node $u$ and are from $\mathcal{D}_i^\tau$ or the root edges of path trees extracted from $\mathcal{G}_i^\circ$. Similarly, for each cycle $c \in \mathcal{G}_i$, there exists a set $\mathbf{C}(c)$ containing all DALG nodes derived from the root edges of path trees extracted from $\mathcal{G}_i^\circ$. By subtracting the set family $\{\mathbf{C}(c) \mid \forall c \in \mathcal{G}_i\}$ from the set family $\{\mathbf{C}(c) \mid \forall c \in \mathcal{G}_i\}$, we can get all DALG nodes belonging to $\mathcal{D}_i^\tau$. Consequently, we can decompose $\mathcal{D}_i$ into $\mathcal{D}_i^\tau$ and $\mathcal{D}_i^\circ$.

Secondly, we prove that $\mathcal{D}_1^\circ$ and $\mathcal{D}_2^\circ$ are isomorphic. Given that $\mathcal{D}_1$ and $\mathcal{D}_2$ are isomorphic, and let $\mathcal{I}^\iota : V(\mathcal{D}_1) \rightarrow V(\mathcal{D}_2)$ be an isomorphism between them. Then, there exists a bijection between the set family $\{\mathbf{D}(u) \mid \forall u \in \mathcal{G}_1\}$ of $\mathcal{D}_1$ and the set family $\{\mathbf{D}(u) \mid \forall u \in \mathcal{G}_2\}$ of $\mathcal{D}_2$. Therefore for each cycle $c$ in $\mathcal{G}_1$, there exists a set $\mathbf{C}(c')$ of $\mathcal{D}_2$, where $c'$ is a cycle from graph $\mathcal{G}_2$, such that $\mathbf{C}(c') \equiv \{\mathcal{I}^\iota(\mathbf{u}) \mid \forall \mathbf{u} \in \mathbf{C}(c)\}$. Hence, the node set that make up $\mathcal{D}_1^\circ$ is bijective with the node set that make up $\mathcal{D}_2^\circ$. Therefore, $\mathcal{D}_1^\circ$ and $\mathcal{D}_2^\circ$ are isomorphic. Consequently, $\mathcal{D}_1^\tau$ and $\mathcal{D}_2^\tau$ are isomorphic.

Since isomorphic graphs always yield isomorphic path trees, we have that $\mathcal{G}_1^\circ$ and $\mathcal{G}_2^\circ$ are isomorphic. Furthermore, $\mathcal{D}_1^\tau$ and $\mathcal{D}_2^\tau$ are generated using the DGL conversion algorithm, and the DLG conversion is reversible. Therefore, $\mathcal{G}_1^\tau$ and $\mathcal{G}_2^\tau$ are isomorphic. If $\mathcal{G}_1^\circ$ and $\mathcal{G}_1^\tau$ have joint nodes, then $\mathcal{G}_2^\circ$ and $\mathcal{G}_2^\tau$ have joint nodes too. Moreover, there exists a bijection between the two joint node sets. Consequently, the combination graph of $\mathcal{G}_1^\circ$ and $\mathcal{G}_1^\tau$, regardless of whether they have joint nodes or not, is isomorphic to the combination graph of $\mathcal{G}_2^\circ$ and $\mathcal{G}_2^\tau$. Hence, $\mathcal{G}_1$ and $\mathcal{G}_2$ are isomorphic.

Consequently, two DALGs are isomorphic if and only if their corresponding original graphs are isomorphic. Thus, the conversion from graphs to DALGs is reversible. □

### C.3 OUR MODELS CAN BE INJECTIVE

We commence the analysis of the injectiveness of our models by reviewing their construction. All of our models adhere to a consistent message passing scheme. Initially, they compute initial node representations as follows:

$$H_{\mathbf{u}}^{(0)} = \Psi(X_u, X_{u,v}, X_v), \tag{1}$$

Subsequently, our models iteratively update node representations on their surrogate structures, DLGs or DALGs, where the $l$-th layer is described as follows:

$$H_{\mathbf{u}}^{(l)} = \Psi^{(l)} \left( H_{\mathbf{u}}^{(0)}, \bigcup{}^{(l)} \left( \left\{ H_{\mathbf{v}}^{(l-1)} \,\middle|\, \mathbf{v} \in \mathcal{N}_{in}(\mathbf{u}) \right\} \right) \right), \tag{2}$$

Afterwards, our models compute the representation of each node $u$ in the original graphs, as follows:

$$Z_u^{\cdot} = \Psi^{\cdot} \left( \bigcup{}^{\cdot} \left( \left\{ \left\{ H_{\mathbf{u}}^{\cdot} \,\middle|\, \mathbf{u} \in \mathbf{D}(u) \right\} \right\} \right) \right), \tag{3}$$

where $H_{\mathbf{u}}^{\cdot}$ could be $H_{\mathbf{u}}^{(L)}$ or $\left[ H_{\mathbf{u}}^{(1)} \parallel H_{\mathbf{u}}^{(2)} \parallel \cdots \parallel H_{\mathbf{u}}^{(L)} \right]$. The concatenation of the node representations across layers, a.k.a Jump Knowledge (Xu et al., 2018), could help to incorporate graph features at different scales. To examine the expressiveness of our models, we focus on $H_u^{\cdot} = H_u^{(L)}$ for each node $u$. This is because the $H_u^{(L)}$ encodes a connected subgraph that is larger than the one encoded by the $H_u^{(l)}$, where $0 \leqslant l < L$. DLGN and ERFGN compute the entire graph's representation $Z_{\mathcal{G}}$ as follows:

$$Z_{\mathcal{G}}^* = \Psi^* \left( \bigcup{}^* \left( \left\{ \left\{ Z_u^{\cdot} \,\middle|\, u \in V \right\} \right\} \right) \right), \tag{4}$$

To model higher-order subgraphs, ERFGN$^\circ$ adds a cycle modeling module to ERFGN, which is described as below:

$$Z_c^\circ = \Psi^\circ \left( \bigcup{}^\circ \left( \left\{ \left\{ H_{\mathbf{u}}^{(L)} \,\middle|\, \mathbf{u} \in \mathbf{C}(c) \right\} \right\} \right) \right). \tag{5}$$

Then, ERFGN$^\circ$ computes the graph representation $Z_{\mathcal{G}}^*$ as follows:

$$Z_{\mathcal{G}}^* = \Psi^* \left( \left[ \bigcup{}^* \left( \left\{ \left\{ Z_u^{\cdot} \,\middle|\, u \in V \right\} \right\} \right) \,\middle\|\, \bigcup{}^* \left( \left\{ \left\{ Z_c^\circ \,\middle|\, c \in C \right\} \right\} \right) \right] \right). \tag{6}$$

To further improve the expressiveness of our models, GloAttNC computes global interactions between a node $u$ to all nodes ($Z_{u,V}^{\cdot\cdot}$), between a node $u$ to all cycles ($Z_{u,C}^{\cdot\circ}$), between a cycle $c$ to all nodes ($Z_{c,V}^{\circ\cdot}$), and between a cycle $c$ to all cycle ($Z_{c,C}^{\circ\circ}$). To cater to different priorities, either efficiency or fine-grained learning, two learning schemes are adopted, i.e., a *1-to-N scheme* and a *1-to-1 scheme*. The *1-to-N scheme* updates these global interactions using Eq. 7, while *1-to-1 scheme* updates these global interactions using Eq. 8.

$$\begin{aligned} Z_{u,V}^{\cdot\cdot} &= \Upsilon^{\cdot\cdot} \left( Z_u^{\cdot}, \bigcup \left( \left\{ \left\{ Z_v^{\cdot} \,\middle|\, v \in V \right\} \right\} \right) \right), \quad & Z_{u,C}^{\cdot\circ} &= \Upsilon^{\cdot\circ} \left( Z_u^{\cdot}, \bigcup \left( \left\{ \left\{ Z_c^\circ \,\middle|\, c \in C \right\} \right\} \right) \right), \\ Z_{c,V}^{\circ\cdot} &= \Upsilon^{\circ\cdot} \left( Z_c^\circ, \bigcup \left( \left\{ \left\{ Z_v^{\cdot} \,\middle|\, v \in V \right\} \right\} \right) \right), \quad & Z_{c,C}^{\circ\circ} &= \Upsilon^{\circ\circ} \left( Z_c^\circ, \bigcup \left( \left\{ \left\{ Z_{c'}^\circ \,\middle|\, c' \in C \right\} \right\} \right) \right). \end{aligned} \tag{7}$$

$$\begin{aligned} Z_{u,V}^{\cdot\cdot} &= \bigcup \left( \left\{ \left\{ \Upsilon^{\cdot\cdot} \left( Z_u^{\cdot}, Z_v^{\cdot} \right) \,\middle|\, v \in V \right\} \right\} \right), \quad & Z_{u,C}^{\cdot\circ} &= \bigcup \left( \left\{ \left\{ \Upsilon^{\cdot\circ} \left( Z_u^{\cdot}, Z_c^\circ \right) \,\middle|\, c \in C \right\} \right\} \right), \\ Z_{c,V}^{\circ\cdot} &= \bigcup \left( \left\{ \left\{ \Upsilon^{\circ\cdot} \left( Z_c^\circ, Z_v^{\cdot} \right) \,\middle|\, v \in V \right\} \right\} \right), \quad & Z_{c,C}^{\circ\circ} &= \bigcup \left( \left\{ \left\{ \Upsilon^{\circ\circ} \left( Z_c^\circ, Z_{c'}^\circ \right) \,\middle|\, c' \in C \right\} \right\} \right). \end{aligned} \tag{8}$$

Subsequently, by using global interactions, the node representations and cycle representations are updated, and then the graph representation is updated as described below:

$$\begin{aligned} \mathbf{Z}_{\mathbf{u}}^{\cdot} &= \Upsilon^{\cdot} \left( Z_u^{\cdot}, Z_{u,V}^{\cdot\cdot}, Z_{u,C}^{\cdot\circ} \right) \\ \mathbf{Z}_{\mathbf{c}}^\circ &= \Upsilon^\circ \left( Z_c^\circ, Z_{c,V}^{\circ\cdot}, Z_{c,C}^{\circ\circ} \right) \\ Z_{\mathcal{G}}^* &= \Upsilon^* \left( \bigcup \left( \left\{ \mathbf{Z}_{\mathbf{u}}^{\cdot} \,\middle|\, u \in V \right\} \right), \bigcup \left( \left\{ \mathbf{Z}_{\mathbf{c}}^\circ \,\middle|\, c \in C \right\} \right) \right), \end{aligned} \tag{9}$$

The provided overview implies that our model incorporates a sequential integration of multiple update functions and aggregators operating on multisets. Consequently, in order to establish the injectiveness of our models, it is essential to demonstrate the injectiveness of each individual component (i.e., an certain update function or a certain message aggregator) as well as the injectiveness of the integration involving all the components. In the following discussion, node representations are simplified to scalars for the sake of clarity, since the same results can be generalised to vector representations.

We first prove that the integration of a simple updating function and a simple message aggregator, i.e., the summation function, can be injective:

**Corollary 1.** *There exists an injective function $\Theta$ that guarantees the existence of an injective message aggregator $\bigcup$ that operates on multisets of the outputs of $\Theta$.*

*Proof.* We first prove that there exists a mapping $\gamma$ so that $\sum_{m \in M} \gamma(m)$ is unique for each multiset $M \subset \mathcal{M}$ of bounded size, where $\mathcal{M} = \left\{ H_{\mathbf{u}}^{(l)} \,\middle|\, \forall \mathbf{u} \right\}$ is the complete multiset of representations, $0 \leqslant l \leqslant L$ and $L$ is the number of layers of our models. Because $\mathcal{M}$ is countable, there exists a mapping $\Gamma : \mathcal{M} \mapsto \mathbb{N}$, where $\mathbb{N}$ is the natural number set. Because the cardinality of multisets $\mathcal{M}$ is bounded, there exists a number $N \in \mathbb{N}$ so that $|M| < N$ for all $M$. Then an example of such $\gamma$ is $\gamma(m) = N^{-\Gamma(m)}$. This $\gamma$ can be viewed as a more compressed form of an one-hot vector or $N$-digit presentation. As a result, the message aggregator $\bigcup$ is an injective function of multisets if it is a summation function such that $\bigcup(\{\gamma(m)|m \in M\}) = \sum_{m \in M} \gamma(m)$. Consequently, such function $\Theta$ exists. $\qquad\square$

We then prove that an update function taking two inputs can be injective.

**Corollary 2.** *There exists an injective function $\Omega : \mathbb{R} \times \mathbb{R} \mapsto \mathbb{R}$, where $\mathbb{R}$ is the rational number set.*

*Proof.* The function $\Omega(x_1, x_2) \equiv x_1 \cdot \lambda + x_2$ is injective for any rational number pair $(x_1, x_2)$, where $\lambda$ is an irrational number. We can prove this by contradiction. Suppose there exists $(x_1', x_2')$ such that $(x_1, x_2) \neq (x_1', x_2')$ but $\Omega(x_1, x_2) = \Omega(x_1', x_2')$ holds. Let's consider two cases: (1) $x_1 = x_1'$ but $x_2 \neq x_2'$, and (2) $x_1 \neq x_1'$. In the first case, if $x_1 = x_1'$ and $\Omega(x_1, x_2) = \Omega(x_1', x_2')$, then it implies that $x_2 = x_2'$. Thus, we reach a contradiction. In the case, we can rewrite $\Omega(x_1, x_2) = \Omega(x_1', x_2')$ as $(x_1 - x_1') \cdot \lambda = x_2 - x_2'$. Since $\lambda$ is an irrational number, the left-hand side of this equation is a non-zero rational number. However, the right-hand side is a rational number. Hence, the equality in this equation cannot hold, and we have reached a contradiction. Therefore, we can conclude that there exists an injective function $\Omega : \mathbb{R} \times \mathbb{R} \mapsto \mathbb{R}$. $\qquad\square$

Please note that the proofs for Corollary 1 and Corollary 2 are adapted from the proofs for Lemma 5 and Lemma 6 in Xu et al. (2019), respectively.

The injectiveness analysis of all components of our models follows:

**Corollary 3.** *All components of each of our models, including all update functions and all message aggregators, can be injective.*

*Proof.* All update functions can be decomposed into a composition of multiple $\Theta$ functions and multiple $\Omega$ functions. For instance, consider the update function for surrogate node initial representation $\Psi$, which can be decomposed into a composition such that $\Psi\left(X_u, X_{u,v}, X_v\right) \equiv \Theta\left(\Omega'\left(\Omega\left(X_u, X_v\right), X_{u,v}\right)\right)$. Suppose that each update function can be decomposed into a composition of multiple $\Theta$ functions and/or multiple $\Omega$ functions, the summation function operating on the outputs of that update function can be an injective message aggregator. $\qquad\square$

**Lemma 8.** *Our models, including DLGN, ERFGN, ERFGN° and ERFGN°+GloAttNC, can be injective if all of its update functions can be decomposed into a composition of multiple $\Theta$ functions and multiple $\Omega$ functions and all its message aggregators are summation operations.*

*Proof.* The composition of injective functions are injective. $\qquad\square$

The analysis presented above established the conditions that guarantee the injectiveness of our models. It is important to emphasize that in order to meet these conditions, each trainable function should be implemented as a Multilayer Perceptron (MLP) comprising at least two linear layers. This requirement is necessary to achieve the universal approximation ability, as outlined in the Universal Approximation Theorem (Hornik et al., 1989).

## C.4 Expressiveness of DLGN, ERFGN, ERFGN° and ERFGN°+GloAttNC

We commence the analysis of the expressiveness of our models by introducing some fundamental definitions in graph theory that are crucial to our study.

**Definition 4** (Eccentricity). *For a connected graph, the **eccentricity** $\epsilon(v)$ of a node $v$ is the greatest distance between $v$ and any other node; in symbols,*

$$\epsilon(v) = \max_{u \in V} \delta(v, u),$$

*where $\delta(v, u)$ is the distance between nodes $u$ and $v$, which is equal to the numbers of edges in a shortest path connecting these nodes.*

**Definition 5** (Radius). *The **radius** $r$ of a connected graph $\mathcal{G}$ is the smallest maximum distance from any node to any other node in the graph, or the minimum eccentricity of any node, in symbols,*

$$r = \min_{v \in V} \max_{u \in V} \delta(v, u) = \min_{v \in V} \epsilon(v).$$

**Definition 6.** *The **diameter** $d$ of a connected graph $\mathcal{G}$ is the largest maximum distance from any node to any other node in the graph, or the maximum eccentricity of any node in the graph, in symbols,*

$$r = \max_{v \in V} \max_{u \in V} \delta(v, u) = \max_{v \in V} \epsilon(v).$$

We also introduce the concept of the *message passing graph*, which is also referred to as the computational graph (Collins, 2018) generated by MPNNs. This concept serves as a framework for understanding the receptive field of our models.

**Message passing graph**   Our proposed model, DLGN or ERFGN, implicitly generates a *message passing graph* from each surrogate structure (DLG or DALG). Assuming the model consists of $L$ layers, where the first $L-1$ layers update surrogate node representations and the last layer updates original node representations, the generated message passing graph consists of $L$ layers too, where the first $L-1$ layers pass message between surrogate nodes, and the $L$-th layer pass message from surrogate nodes to original nodes. Consequently, each path in the message passing graph, a.k.a *message passing path*, which extends from an input node to an output node, corresponds to a walk (sequence of connected nodes and edges) in the original graph with length up to $L$.

We proceed to analyze the requirements concerning the receptive field necessary for effectively express non-isomorphic graphs.

**Lemma 9.** *The representation of an original node encodes the information of the subtree with that node as its root and a height of $l$, where $l \leqslant L$.*

*Proof.* The structure of the message passing graph ensures that an original node can receive information from other nodes within a maximum distance of $L$ in the original graph. □

**Corollary 4.** *For any connected graph $\mathcal{G}$ with a radius $r$ and a diameter $d$, it holds that $d \leqslant 2r$.*

*Proof.* Take nodes $u, v \in \mathcal{G}$ such that $\delta(u, v) = d$ and let $o$ be a central node in $\mathcal{G}$ such that $\epsilon(o) = r$. Thus, $\delta(o, t) \leqslant r, \forall t \in G$. By triangle inequality, $d = \delta(u, v) \leqslant \delta(u, o) + \delta(o, v) \leqslant 2r$. □

The lemma and corollary above suggest that if the largest radius of a set of non-isomorphic connected graphs is $R$, then the minimum number of layers of our models DLG and ERFGN is $R$.

**Corollary 5.** *For two non-isomorphic graphs $\mathcal{G}_1$ and $\mathcal{G}_2$ with respective diameters $d_1$ and $d_2$, where $d_1 \geqslant d_2$, which are both connected, there exists at least one path in $\mathcal{G}_1$ of length $l > d_2$ that is not present in $\mathcal{G}_2$.*

*Proof.* The length of each path in $\mathcal{G}_2$ does not exceed $d_2$. □

**Corollary 6.** *For two non-isomorphic and connected graphs $\mathcal{G}_1$ and $\mathcal{G}_2$, with diameters $d_1$ and $d_2$ and radii $r_1$ and $r_2$ respectively, where $r_1 \geqslant r_2$, there exists at least one subgraph of radius $r_1 > r_2$ in $\mathcal{G}_1$ that is not present in $\mathcal{G}_2$.*

*Proof.* Any path in $\mathcal{G}_1$ of length $l$, where $l > 2r_2 \geqslant d_2$, is not present in $\mathcal{G}_2$. The subgraph formed by this path, which has a radius of $\lceil l/2 \rceil$, where $l/2 > r_2$, is not present in $\mathcal{G}_2$. □

The above lemma and corollaries suggest that DLGN or ERFGN can discriminate non-isomorphic connected graphs if the number layer of DLGN or ERFGN is not less than the largest radius of these connected graphs, and the model does not suffer from the message confusion problem.

**Definition 7.** *A component of a graph is a connected subgraph that is not part of any larger connected subgraph.*

**Corollary 7.** *For non-isomorphic graphs, no matter how connected or disconnected, let the largest radius of the components included in these graphs is $R$, each graph contains at least one subgraph of radius $r \leqslant R$ that does not present in others.*

*Proof.* The proof of this corollary follows a similar approach to the proof of Corollary 6. □

**Corollary 8.** *Both DLGN and ERFGN, when composed of $L$ layers, can effectively express a set of non-isomorphic graphs if the maximum radius $R$ of the components contained in the graph set satisfies $R \leqslant L$ and if the models do not suffer from the message confusion problem.*

*Proof.* When the maximum radius $R$ of the components contained in the graph set satisfies $R \leqslant L$, DLGN and ERFGN with $L$ layers can produce node representations for any graph, which together encode all the paths in the graph. When the models do not suffer from the message confusion problem, they can learn the information of all the paths in the graph and thus the information all subgraphs. Therefore, they can produce identifiable graph representations for these non-isomorphic graphs. □

However, DLGN may suffer from the *message confusion* problem, when input graphs include cycles. This is because DLGN may generate the identical message passing paths from cycles and paths present in original graphs. To illustrate this, let's consider the example of a cycle $(a, b, c, d)$ and a path $a \rightarrow b \rightarrow c \rightarrow d \rightarrow a$, where $a, b, c, d$ represent node labels. In this scenario, DLGN can produce the same message passing paths, denoted as $\mathbf{a} \rightarrow \mathbf{b} \rightarrow \mathbf{c} \rightarrow \mathbf{d}$, where $\mathbf{a}, \mathbf{b}, \mathbf{c}, \mathbf{d}$ correspond to DLG nodes derived from the edges $a \rightarrow b$, $b \rightarrow c$, $c \rightarrow d$, and $d \rightarrow a$, respectively.

**Lemma 10.** *DLGN, when composed of $L$ layers, successfully avoids the problem of message confusion when confronted with input graphs that are cycle-free or contain only cycles larger than $L$.*

*Proof.* When input graphs that are cycle-free or contain only cycles larger than $L$, the phenomenon that cycles and paths in surrogate structures generate identical message passing paths cannot happen. □

**Lemma 3.** *DLGN with $L$ layers can express all subtrees of non-isomorphic graphs if the minimum cycle size $S$ and the maximum component radius $R$ of these graphs satisfy $R \leqslant L < S$.*

*Proof.* When the given non-isomorphic graphs have a minimum cycle size $S \geqslant L$, DLGN with $L$ layers does not suffer from the message confusion problem. Therefore, DLGN with $L$ layers can produce different node representations for non-isomorphic connected subgraphs with radius up to $L$. When the largest component-maximum-radius among these graphs $R$ satisfies $R \leqslant L$, each graph contains at least one connected subgraph with radius up to $L$ that does not present in the others. Therefore, DLGN with $L$ layers can produce different graph representations for these non-isomorphic graphs. □

**Lemma 4.** *ERFGN with $L$ layers can express all subtrees of non-isomorphic graphs if the maximum component radius $R$ of these graphs satisfies $R \leqslant L$.*

*Proof.* When the largest component-maximum-radius among these graphs is up to $L$, each graph contains at least one connected subgraph with radius up to $L$ that does not present in the others. ERFGN with $L$ layers does not suffer from the problem of message confusion, thus it can produce different node representations for non-isomorphic connected subgraphs with radius up to $L$. Consequently, ERFGN with $L$ layers can produce different graph representations for these non-isomorphic graphs. □

**Lemma 5.** *ERFGN° can express all connected subgraph with a cycle.*

*Proof.* ERFGN° updates the representation of each cycle by aggregating the representations of the DALG nodes that belong to that cycle. Therefore, the representation of that cycle encodes not only the cycle itself, but also all the subtrees connected to that cycle. Hence, the retention of topological information within subtrees extracted from nodes in a cycle enables the representation of the cycle itself. Graph readouts, which aggregate information from all extracted subtrees, have demonstrated the ability to express non-isomorphic graphs effectively. Hence, a cycle aggregator can be considered equivalent to a graph readout, such as a summation function, to achieve the comprehensive expressiveness of cycles. Moreover, the trainability of cycle aggregators can be established using similar approaches as those employed for proving the trainability of graph readouts. □

ERFGN lacks the ability to generate identifiable representations for nodes. This limitation is due to the fact that ERFGN updates a representation for a node using only the representations of all paths ending at that node, losing information about any cycles connected to that node. Consequently, ERFGN would produce indistinguishable representations for nodes within non-isomorphic graphs. Similarly, ERFGN° lacks the ability to produce identifiable representations for nodes and cycles. This is because ERFGN° cannot model the connection between a subtree and multiple subcycles.

**Lemma 6.** *ERFGN°+GloAttNC can generate identifiable representations for nodes and cycles.*

*Proof.* Every graph possesses at least one path that cannot be found in any non-isomorphic graph. ERFGN has the ability to generate an identifiable representation for each node that encodes all paths ending at that particular node. As a result, the aggregations of node representations for non-isomorphic graphs can differ. ERFGN°+GloAttNC updates a new representation for each node using the aggregation of all node representations. Therefore, the updated representation can include the isomorphism information of the graph containing the node. Consequently, ERFGN°+GloAttNC has the ability to generate identifiable node representations if ERFGN can express all subtrees and ERFGN°+GloAttNC is injective. Similarly, ERFGN°+GloAttNC can generate identifiable cycle representations under the same conditions. □

## C.5 MODEL RANKING OF SUBGRAPH EXPRESSIVITY

In this work, we consider that the subgraph expressivity of a model quantifies how many non-isomorphic subgraphs this model can distinguish. Before the expressivity comparison, it is crucial to review how other powerful MPNNs express subgraphs. Typical MPNNs, represented by the Graph Isomorphism Network (GIN) (Xu et al., 2019), are the cornerstone of all robust MPNNs. Then, with $H_u^{(l)}$ denoting the representation of node $u$ in layer $l$, $H_u^{(l)}$ is updated as follows:

$$H_u^{(l)} = \Psi^{(l)} \left( H_u^{(l-1)}, \bigcup{}^{(l)} \left( \left\{ \left\{ H_v^{(l-1)} \mid v \in \mathcal{N}(u) \right\} \right\} \right) \right), \tag{10}$$

where $\mathcal{N}(u)$ is the neighboring nodes of node $u$, and $\bigcup^{(l)}$ is a message aggregator. Xu et al. (2019) have stated that GIN provably generalizes the 1-WL test (Leman & Weisfeiler, 1968) and the WL Subtree kernel (Shervashidze et al., 2011). Both the 1-WL test (Leman & Weisfeiler, 1968) and typical MPNNs adopt a iterative paradigm to update a node by aggregating the neighbors of the node. The iterative paradigm defines a tree structure called a WL subtree, in which a node connects to its parent node if they are adjacent in the source graph. Consequently, the 1-WL test (Leman & Weisfeiler, 1968) and typical MPNNs can only express WL subtrees extracted from source graphs. However, the *message passing redundancy* issue can lead to non-isomorphic cyclic graphs generate isomorphic WL subtrees (Chen et al., 2022). As a result, typical MPNNs can not express arbitrary subtrees. As our DLGN model can not address *message passing redundancy* which arises from subcycles, DLGN can only express WL subtrees too.

RFGNN (Chen et al., 2022) extracts *epath* trees (a.k.a TPTs) from source graphs. An *epath* tree extracted for a node from a graph consists of that node, 1-hop neighboring nodes, 2-hop neighboring nodes and so on, where a $(l)$-hop neighbor connects a $(l-1)$-hop neighbor if they are adjacent in the source graph. In addition, an *epath* in such tree allows its start and end nodes to be the same source node, in order to represent a cycle. The utilizing of the *epath* tree structure avoid the *message passing redundancy* problem. As a result, RFGNN (Chen et al., 2022) can express arbitrary subtrees.

For example, an epath extracted from a node in a cycle of size 3 can also be extracted from the centre node of a path of length 6. As a result, RFGNN (Chen et al., 2022) does not perform well

at capturing cycles. Contrary, ERFGN$^\circ$ explicitly models cycles and achieves the expressivity of subgraphs with a cycle.

$k$-GNNs (Bodnar et al., 2021a) is the representive models of higher-order MPNNs using $k$-WL. Instead of updating node representations, $k$-GNNs consider a GNN that corresponds to a set version of a $k$-WL algorithm. Specifically, for any set $\underline{V} \subseteq V$ with $|\underline{V}| = k$, let $\mathcal{N}^k(\underline{V}) = \{V' \subset V, |V'| = k$ and $|V' \cap \underline{V}| = k-1\}$. $k$-GNNs (Bodnar et al., 2021a) then updates as

$$H_{\underline{V}}^{(l)} = \Psi^{(l)}\left(H_{\underline{V}}^{(l-1)}, \sum_{V' \in \mathcal{N}^k(\underline{V})} H_{V'}^{(l-1)}\right) \tag{11}$$

$k$-GNNs achieves a expressivity related to $k$-WL, which is strictly more powerful than 1-WL when $k \geqslant 3$. However, Bodnar et al. (2021a) have proven that 3-WL cannot count chordless cycles of size strictly larger than 3. Consequently, ERFGN$^\circ$ is strictly more powerful than $k$-GNNs using 3-WL.

ERFGN updates a representation for a node using only the representations of all paths ending at that node, losing information about any cycles connected to that node. ERFGN$^\circ$ cannot model the connection between a subtree and multiple subcycles. ERFGN$^\circ$+GloAttNC can achieve higher expressiveness than ERFGN and ERFGN$^\circ$ because ERFGN$^\circ$+GloAttNC can generate identifiable node/cycle representations for nodes/cycles located on non-isomorphic subgraphs.

**Lemma 7.** *Subgraph expressiveness can be ranked as follows: typical MPNNs = DLGN < ERFGN$\simeq$ RFGNN (Chen et al., 2022) < higher-order MPNNs using 3-WL (Bodnar et al., 2021a) < ERFGN$^\circ$< ERFGN$^\circ$+GloAttNC.*

*Proof.* Based on the discussion above, we can conclude this expressivity ranking. $\square$

## D    DETAILS OF DATASETS

Table 7: Details of Datasets

| Dataset | # Graphs | Avg. # nodes | Avg. # edges | Task type | Metric |
|---|---|---|---|---|---|
| MNIST | 70,000 | 70.6 | 564.5 | 10-class classif. | Accuracy |
| CIFAR10 | 60,000 | 117.6 | 941.1 | 10-class classif. | Accuracy |
| Peptides-func | 15,535 | 150.9 | 307.3 | 10-task classif. | Avg. Precision |
| Peptides-struct | 15,535 | 150.9 | 307.3 | 11-task regression | Mean Abs. Error |
| ZINC subset | 12,000 | 23.2 | 24.9 | regression | Mean Abs. Error |
| OGBG-MOLHIV | 41,127 | 25.5 | 27.5 | binary classif. | AUROC |
| QM9 | 130,831 | 18.0 | 37.3 | 12-task regression | Mean Abs. Error |
| ENZYMES | 600 | 32.63 | 62.14 | 6-class classif. | Accuracy |
| PROTEINS_full | 1113 | 39.01 | 72.82 | binary classif. | Accuracy |
| FRANKENSTEIN | 4,337 | 16.90 | 17.88 | binary classif. | Accuracy |
| NCI1 | 4,100 | 29.87 | 32.30 | binary classif. | Accuracy |

The datasets used in our experiments are described in detail in Table 7.

**MNIST and CIFAR10** (Dwivedi et al., 2023) are derived from like-named image classification datasets by constructing an 8 nearest-neighbor graph of SLIC superpixels for each image. The 10-class classification tasks and standard dataset splits follow the original image classification datasets, i.e., for MNIST 55K/5K/10K and for CIFAR10 45K/5K/10K train/validation/test graphs.

**Peptides-func** and **Peptides-struct** datasets from Long-Range Graph Benchmarks (Dwivedi et al., 2022) are specifically designed to test the capability of models in performing tasks that require reasoning with long-range interactions. Peptides-func and Peptides-struct (Dwivedi et al., 2022) are both composed of atomic graphs of peptides retrieved from SATPdb (Singh et al., 2016). In Peptides-func, the prediction task is multi-label graph classification into 10 nonexclusive peptide functional classes. For Peptides-struct, the task is graph regression of 11 3D structural properties of

the peptides. Stratified splitting was applied to generate balanced train-valid-test dataset splits in the ratio of 70%-15%-15% (Dwivedi et al., 2022).

**QM9** (Ramakrishnan et al., 2014; Wu et al., 2018) is a graph dataset with 12 regression tasks. It contains 130K small molecules, which are divided into train-valid-test sets at a ratio of 80%-10%-10%. The task here is to separately perform regressions on 12 targets representing energetic, electronic, geometric, and thermodynamic properties, based on the graph structure and node/edge features. All evaluation metrics are mean absolute error (MAE).

**ZINC Subset** (Dwivedi et al., 2023) consists of 12K molecular graphs from the ZINC database of commercially available chemical compounds. These molecular graphs are between 9 and 37 nodes large. Each node represents a heavy atom (28 possible atom types) and each edge represents a bond (3 possible types). The task is to regress a molecular property known as the constrained solubility which is the term logP-SA-cycle (octanol-water partition coefficients, logP, penalized by the synthetic accessibility score, SA, and number of long cycles, cycle). The dataset comes with a predefined 10K/1K/1K train/validation/test split.

**OGBG-MOLHIV** is a molecular property prediction dataset adopted by OGB (Hu et al., 2020) from MoleculeNet (Wu et al., 2018). The dataset use a common node (atom) and edge (bond) featurization that represent chemophysical properties. The prediction task is binary classification of molecule's fitness to inhibit HIV replication. We used the provided splits of OGB Hu et al. (2020).

**TUDatasets** is a collection of benchmark datasets commonly used for the evaluation of GNNs (Morris et al., 2020a). We evaluate our proposed models on six real-world datasets from this collection. Specifically, ENZYMES (Borgwardt et al., 2005; Schomburg et al., 2004) and PRO-TEINS_full (Borgwardt et al., 2005; Dobson & Doig, 2003) are a medium-size bio-informatics dataset, FRANKENSTEIN Orsini et al. (2015), NCI1 (Wale et al., 2008; Kriege & Mutzel, 2012) and NCI109 (Wale et al., 2008; Kriege & Mutzel, 2012) are small-size molecular datasets.

## E  COMPUTING ENVIRONMENT AND USED RESOURCES

Our code implementation is based on PyG Fey & Lenssen (2019)[6] and its modules torch-scatter and torch-sparse. All experiments were run on a Linux platform with $4 \times$ NVidia GeForce RTX 3090 GPUs (24GB) and $1 \times$ AMD EPYC 75F3 32-Core Processor. The resource budget for each experiment was 1 GPU and 6 process cores.

## F  DETAILS OF EXPERIMENTS

To validate our proposed cycle modeling module and GloAttNC, we propose an ablation module, called GloAtt, which only learns the interactions between node-to-node pairs. It computes global attention using Eq. 12:

$$
Z_{u,V}^{\cdot\cdot} = \begin{cases} \Upsilon^{\cdot\cdot}\left(Z_u^{\cdot}, \bigcup\left(\left\{\left\{Z_v^{\cdot} \mid v \in V\right\}\right\}\right)\right), & \text{1-to-}N \\ \bigcup\left(\left\{\left\{\Upsilon^{\cdot\cdot}(Z_u^{\cdot}, Z_v^{\cdot}) \mid v \in V\right\}\right\}\right), & \text{1-to-1} \end{cases} \tag{12}
$$

Subsequently, the representations of nodes and graphs are computed sequentially:

$$
\begin{aligned}
\mathbf{Z}_u^{\cdot} &= \Upsilon^{\cdot}\left(Z_u^{\cdot}, Z_{u,V}^{\cdot\cdot}\right), \\
Z_{\mathcal{G}}^* &= \begin{cases} \Upsilon^*\left(\bigcup\left(\left\{\mathbf{Z}_u^{\cdot} \mid u \in V\right\}\right)\right), & \text{without cycle modeling,} \\ \Upsilon^*\left(\bigcup\left(\left\{\mathbf{Z}_u^{\cdot} \mid u \in V\right\}\right), \bigcup\left(\left\{Z_c^{\circ} \mid c \in C\right\}\right)\right), & \text{with cycle modeling.} \end{cases}
\end{aligned} \tag{13}
$$

### F.1  MNIST AND CIFAR10

For both MNIST and CIFAR10 datasets (Dwivedi et al., 2023), we only utilize a 3-layer DLGN. The pre-computation times for converting the source graphs of MNIST and CIFAR10 into DLGs

---

[6]https://pyg.org/

Table 8: Ablation study and hyper-parameters tuning of DLGN on MNIST and CIFAR10.

| Dataset | Graph Readout | GloAtt | Accuracy ( ($\uparrow$) ) | # Parameters | Training Time (epoch/total) | best epoch |
|---------|---------------|--------|---------------------------|--------------|-----------------------------|------------|
| MNIST | mean | no | 98.630$\pm$0.034 | 107,434 | 39.5s / 1.1h | 89.0 |
| MNIST | mean | 1N | 98.608$\pm$0.119 | 131,158 | 40.0s / 1.1h | 96.2 |
| MNIST | sum | no | **98.640$\pm$0.052** | 107,434 | 39.5s / 1.1h | 92.6 |
| MNIST | sum | 1N | 98.606$\pm$0.055 | 131,158 | 40.0s / 1.1h | 89.6 |
| CIFAR10 | mean | no | **73.386$\pm$0.312** | 107,578 | 48.2s / 1.3h | 52.2 |
| CIFAR10 | mean | 1N | 73.194$\pm$0.713 | 131,302 | 49.7s / 1.4h | 46.8 |
| CIFAR10 | sum | no | 72.950$\pm$0.595 | 107,578 | 48.2s / 1.3h | 54.0 |
| CIFAR10 | sum | 1N | 73.268$\pm$0.375 | 131,302 | 49.7s / 1.4h | 47.8 |

are **41.43s** and **65.15s**, respectively. Furthermore, we utilize the CrossEntropyLoss function and the AdamW (Loshchilov & Hutter, 2018) optimizer with weight decay of $10^{-5}$ to train our models. Additionally, the batch size is 32, and the learning rate is $10^{-3}$, which is halved every 50 epochs before falling below $10^{-5}$. The number of warm-up epochs is 5. The maximum number of epochs is 150, but training will stop early after 100 epochs if performance on the validation set does not improve for 50 epochs. We validate DLGN and DLGN+GloAtt with a *sum* or *mean* graph readout. The results are shown in Table 8, where the 'GloAtt' column indicates different global attention modeling schemes, where 'no' means turning off global attention modeling, '1N' means modeling node-to-node global attention using the '1-to-N' scheme. Such notation is also used in other tables.

### F.2    PEPTIDES-FUNC AND PEPTIDES-STRUCT

Table 9: Statistical data of Peptides graph conversions. (R: maximum cycle size. H: maximum tree height.)

| | source | DLG | DALG | | |
|---|--------|-----|------|------|------|
| | | | R6H4 | R6H6 | R10H10 |
| Time | - | 3.17s | 36.21s | 38.38s | 38.69s |
| Avg. # nodes | 150.9 | 307.3 | 390.8 | 470.5 | 504.4 |
| Avg. # edges | 307.3 | 411.5 | 474.1 | 569.2 | 607.5 |
| Avg. # cycles | - | - | 3.5 | 3.5 | 3.5 |

We validate all our models on both Peptides-func and Peptides-struct datasets (Dwivedi et al., 2022). We utilize the CrossEntropyLoss function for Peptides-func dataset, and the L1Loss function for Peptides-struct dataset. Furthermore, we use the AdamW (Loshchilov & Hutter, 2018) optimizer with weight decay of $10^{-5}$ to train our models. Additionally, the batch size is 32, and the learning rate is $10^{-3}$, which is halved every 50 epochs before falling below $10^{-5}$. The number of warm-up epochs is 5. The maximum number of epochs is 150, but training will stop early after 100 epochs if performance on the validation set does not improve for 50 epochs.

We illustrate the pre-computation process of converting the source graphs into DLGs/DALGs. Since both datasets consist of the same graphs, the pre-computation time and results are identical, as shown in Table 9. The statistical data reveal the high efficiency of our methods in constructing surrogate graphs. Even tasks such as extracting cycles and path trees of maximum 10 are completed quickly, without generating DALGs of large size.

Subsequently, we conduct an ablation study and hyper-parameters tuning experiments for DLGN on Peptides-struct and Peptides-func datasets (Dwivedi et al., 2022), separately. Additionally, we measure the training efficiency of our models. As shown in Table 10 and Table 11, DLGN demonstrates high training efficiency.

Finally, we perform ablation studies and hyper-parameter tuning experiments for ERFGN on Peptides-struct and Peptides-func datasets (Dwivedi et al., 2022). These experiments can serve to validate the effectiveness of our proposed cycle modeling module and global attention module (GloAttNC) to improve the performance of our ERFGN model. The experiments results are shown

Table 10: Ablation study and hyper-parameters tuning of DLGN on Peptides-func.

| GloAtt | # Layers | Test AP ( ($\uparrow$) ) | # Parameters | Training time (epoch/total) | best epoch |
|--------|----------|--------------------------|--------------|------------------------------|------------|
| no | 4 | $0.6591 \pm 0.0046$ | 358,811 | 4.4s/7.4m | 39.2 |
| no | 6 | $0.6680 \pm 0.0078$ | 389,995 | 4.5s/7.5m | 44.6 |
| no | 8 | $\mathbf{0.6764 \pm 0.0055}$ | 403,323 | 4.6s/7.6m | 51.8 |
| 1N | 4 | $0.6633 \pm 0.0072$ | 424,271 | 4.4s/7.4m | 59.0 |
| 1N | 6 | $0.6657 \pm 0.0049$ | 447,059 | 4.5s/7.5m | 59.6 |
| 11 | 4 | $0.6473 \pm 0.0044$ | 402,731 | 4.4s/7.4m | 54.2 |
| 11 | 6 | $0.6519 \pm 0.0054$ | 428,299 | 4.5s/7.5m | 54.6 |

Table 11: Ablation study and hyper-parameters tuning of DLGN on Peptides-struct.

| GloAtt | # Layers | Test MAE ($\downarrow$) | # Parameters | Training time (epoch/total) | best epoch |
|--------|----------|--------------------------|--------------|------------------------------|------------|
| no | 4 | $0.2568 \pm 0.0015$ | 358,811 | 3.7s/6.1m | 39.2 |
| no | 6 | $0.2568 \pm 0.0021$ | 389,995 | 4.3s/7.2m | 44.6 |
| no | 8 | $0.2540 \pm 0.0008$ | 403,323 | 4.9s/8.1m | 51.8 |
| 1N | 4 | $\mathbf{0.2483 \pm 0.0011}$ | 424,271 | 4.1s/6.8m | 59.0 |
| 1N | 6 | $0.2485 \pm 0.0022$ | 447,059 | 4.5s/7.6m | 59.6 |
| 11 | 4 | $0.2524 \pm 0.0013$ | 402,731 | 13.6s/22.6m | 54.2 |
| 11 | 6 | $0.2525 \pm 0.0019$ | 428,299 | 13.8s/23.0m | 54.6 |

Table 12: Ablation study and hyper-parameters tuning on Peptides-func. The best results of ERFGN, ERFGN° and ERFGN°+GloAttNC are colored in **blue**, **green** and **red**, respectively.

| GloAttNC | Model Cycle | TreeHeight /CycleSize | Test AP ($\uparrow$) | # Parameters | Training Time (epoch/total) | best epoch |
|----------|-------------|------------------------|----------------------|--------------|------------------------------|------------|
| no | no | 4/6 | $0.6688 \pm 0.0066$ | 358,690 | 4.0s/6.6m | 65 |
| no | no | 6/6 | $0.6766 \pm 0.0032$ | 389,882 | 4.7s/7.9m | 61 |
| no | no | 10/10 | $\mathbf{0.6790 \pm 0.0055}$ | 401,002 | 6.1s/10.1m | 76.6 |
| no | yes | 4/6 | $0.6609 \pm 0.0062$ | 209,592 | 4.5s/7.5m | 69.6 |
| no | yes | 6/6 | $0.6650 \pm 0.0006$ | 255,000 | 5.3s/8.9m | 63.5 |
| no | yes | 10/10 | $\mathbf{0.6869 \pm 0.0056}$ | 345,816 | 6.5s/10.9m | 63.2 |
| 1N | no | 4/6 | $0.6741 \pm 0.0042$ | 424,150 | 4.4s/7.4m | 75.8 |
| 1N | no | 6/6 | $0.6810 \pm 0.0077$ | 446,946 | 4.9s/8.1m | 72 |
| 1N | no | 10/10 | $0.6779 \pm 0.0070$ | 443,002 | 6.2s/10.3m | 62.6 |
| 1NC | yes | 4/6 | $0.6882 \pm 0.0098$ | 299,634 | 5.4s/9.0m | 74.4 |
| 1NC | yes | 6/6 | $\mathbf{0.6912 \pm 0.0049}$ | 435,858 | 6.0s/9.9m | 58.4 |
| 1NC | yes | 10/10 | $0.6814 \pm 0.0037$ | 435,858 | 7.2s/12.0m | 75.6 |
| 11 | no | 4/6 | $0.6527 \pm 0.0030$ | 402,610 | 13.9s/23.2m | 66.8 |
| 11 | no | 6/6 | $0.6576 \pm 0.0088$ | 428,186 | 14.6s/24.4m | 68 |
| 11 | no | 10/10 | $0.6618 \pm 0.0066$ | 429,226 | 15.9s/26.5m | 85.2 |
| 11C | yes | 4/6 | $0.6855 \pm 0.0065$ | 390,450 | 14.5s/24.2m | 58.2 |
| 11C | yes | 6/6 | $0.6818 \pm 0.0042$ | 345,042 | 15.3s/25.4m | 88 |
| 11C | yes | 10/10 | $0.6832 \pm 0.0078$ | 526,674 | 16.6s/27.7m | 63.6 |

in Table 12 and Table 13, respectively, where the 'GloAttNC' column indicates different global attention modeling schemes, where 'no' means the disable of global attention modeling, '1N' means the modeling of node-to-node global attention using the '1-to-N' scheme, '11' means the using of the '1-to-1' scheme, and '1NC' and '11C' means the modeling of global attention of nodes and cycles. Such notation is used in the results of all ablation studies and hyper-parameter tuning. The shown test AP/MAE is the mean $\pm$ s.d. of results from 5 runs. The model layers tested are 4, 6, and 10. From the results, the following conclusions can be drawn: Layers above 6 could not provide better performance on Peptides-func, while layers above 4 could not provide better performance

Table 13: Ablation study and hyper-parameters tuning on Peptides-struct. The best results of ER-FGN, ERFGN° and ERFGN°+GloAttNC are colored in **blue**, **green** and **red**, respectively.

| GloAttNC | Model Cycle | TreeHeight /CycleSize | Test MAE (↓) | # Parameters | Training Time (epoch/total) | best epoch |
|---|---|---|---|---|---|---|
| no | no | 4/6 | **0.2553 ± 0.0028** | 358,690 | 4.0s/6.6m | 65 |
| no | no | 6/6 | 0.2568 ± 0.0021 | 389,882 | 4.7s/7.9m | 61 |
| no | no | 10/10 | 0.2562 ± 0.0015 | 401,002 | 6.1s/10.1m | 76.6 |
| no | yes | 4/6 | **0.2563 ± 0.0011** | 209,592 | 4.5s/7.5m | 69.6 |
| no | yes | 6/6 | 0.2582 ± 0.0020 | 255,000 | 5.3s/8.9m | 63.5 |
| no | yes | 10/10 | 0.2565 ± 0.0019 | 345,816 | 6.5s/10.9m | 63.2 |
| 1N | no | 4/6 | 0.2483 ± 0.0013 | 424,150 | 4.4s/7.4m | 75.8 |
| 1N | no | 6/6 | 0.2485 ± 0.0022 | 446,946 | 4.9s/8.1m | 72 |
| 1N | no | 10/10 | 0.2473 ± 0.0011 | 443,002 | 6.2s/10.3m | 62.6 |
| 1NC | yes | 4/6 | **0.2468 ± 0.0014** | 299,634 | 5.4s/9.0m | 74.4 |
| 1NC | yes | 6/6 | 0.2476 ± 0.0010 | 435,858 | 6.0s/9.9m | 58.4 |
| 1NC | yes | 10/10 | 0.2481 ± 0.0008 | 435,858 | 7.2s/12.0m | 75.6 |
| 11 | no | 4/6 | 0.2537 ± 0.0031 | 402,610 | 13.9s/23.2m | 66.8 |
| 11 | no | 6/6 | 0.2525 ± 0.0019 | 428,186 | 14.6s/24.4m | 68 |
| 11 | no | 10/10 | 0.2527 ± 0.0024 | 429,226 | 15.9s/26.5m | 85.2 |
| 11C | yes | 4/6 | 0.2477 ± 0.0014 | 390,450 | 14.5s/24.2m | 58.2 |
| 11C | yes | 6/6 | 0.2480 ± 0.0021 | 345,042 | 15.3s/25.4m | 88 |
| 11C | yes | 10/10 | 0.2480 ± 0.0022 | 526,674 | 16.6s/27.7m | 63.6 |

on Peptides-struct. The inclusion of cycle modeling in GloAttNC gives the best performance, but it appears that cycle modeling alone may not provide substantial improvements. In addition, the statistical training times show that our models are efficient at solving tasks at Peptides-struct and Peptides-func datasets (Dwivedi et al., 2022).

## F.3 ZINC SUBSET

Table 14: Statistical data of ZINC Subset graph conversions.

| | Time | Avg. # nodes | Avg. # edges | Avg. # cycles |
|---|---|---|---|---|
| source | - | 23.2 | 49.9 | - |
| DLAG of R8H4 | 15.29s | 118.6 | 126.5 | 31.3 |

Table 15: Ablation study on ZINC Subset dataset.

| Scheme | no | 1NC | 11C |
|---|---|---|---|
| Test MAE (↓) | 0.106±0.004 | 0.0747±0.004 | **0.0684±0.002** |
| Training Time (epoch/total) | 5.2s/2.9h | 5.7s/3.2h | 6.0s/3.3h |

As the task of the ZINC Subset (Dwivedi et al., 2023) is to regress a molecular property related to cycles, we only apply our models with cycle modeling, i.e., ERFGN° and ERFGN°+GloAttNC. We construct DALGs by extracting cycles with a maximum cycle size of 8 and path trees with a maximum height 4. The pre-computation time and results are as shown in Table 14. Additionally, the function is L1Loss, the optimizer is AdamW (Loshchilov & Hutter, 2018) with weight decay of $10^{-5}$, the batch size is 64, and the learning rate is $10^{-3}$, which is halved every 200 epochs before falling below $10^{-5}$. Following with (Rampášek et al., 2022), the number of warm-up epochs is 50, the number of epochs is 2000. The experiments results are shown in Table 15. The tow tables support that our models can achieve high efficiency and high performance on graph set with abundant cycles.

## F.4 QM9

Table 16: Statistical data of QM9 graph conversions.

|  | Time | Avg. # nodes | Avg. # edges | Avg. # cycles |
|---|---|---|---|---|
| source | - | 18.0 | 37.3 | - |
| DLAG of R8H6 | 147.1s | 110.4 | 211.1 | 1.9 |

As for QM9 (Ramakrishnan et al., 2014; Wu et al., 2018), we only apply our ERFGN$^\circ$+GloAttNC model. We construct DALGs by extracting cycles with a maximum cycle size of 8 and path trees with a maximum height 6. The pre-computation time and results are as shown in Table 16. Additionally, the function is L1Loss, the optimizer is AdamW (Loshchilov & Hutter, 2018) with weight decay of $10^{-5}$, the batch size is 256, and the learning rate is $10^{-3}$, halved every 50 epochs before falling below $10^{-5}$. The maximum number of epochs is 350, but training will stop early after 200 epochs if performance on the validation set does not improve.

## F.5 TUDATASETS

Table 17: Ablation study and hyper-parameters tuning on TUDatasets.

| Dataset | TreeHeight /CycleSize | GloAtt | Accuracy (↑) | # Parameters | Best epoch |
|---|---|---|---|---|---|
| ENZYMES | 3 / 6 | no | 73.500±5.890 | 93,894 | 45.1 |
| ENZYMES | 4 / 6 | no | 71.500±6.121 | 106,566 | 43.8 |
| ENZYMES | 3 / 6 | 1N | 73.333±6.455 | 112,678 | 38.7 |
| ENZYMES | 4 / 6 | 1N | 74.500±5.273 | 125,350 | 58.8 |
| ENZYMES | 3 / 6 | 1NC | 74.833±5.134 | 144,006 | 57.7 |
| ENZYMES | 4 / 6 | 1NC | 72.167±6.710 | 156,678 | 42.3 |
| PROTEINS_full | 3 / 6 | no | 77.538±3.295 | 93,921 | 20.7 |
| PROTEINS_full | 4 / 6 | no | 77.539±3.483 | 106,593 | 27.9 |
| PROTEINS_full | 3 / 6 | 1N | 77.267±3.356 | 112,705 | 25.2 |
| PROTEINS_full | 4 / 6 | 1N | 77.087±3.310 | 125,377 | 30.9 |
| PROTEINS_full | 3 / 6 | 1NC | 78.253±3.360 | 144,033 | 55.5 |
| PROTEINS_full | 4 / 6 | 1NC | 78.432±3.880 | 156,705 | 49.4 |
| FRANKENSTEIN | 3 / 6 | no | 76.965±1.247 | 142,849 | 59.5 |
| FRANKENSTEIN | 4 / 6 | no | 77.196±1.704 | 155,521 | 56.7 |
| FRANKENSTEIN | 5 / 6 | no | 76.989±1.445 | 168,193 | 62.8 |
| FRANKENSTEIN | 6 / 6 | no | 76.250±1.161 | 180,865 | 35.1 |
| FRANKENSTEIN | 3 / 6 | 1N | 77.104±1.735 | 161,633 | 29.9 |
| FRANKENSTEIN | 4 / 6 | 1N | 77.496±1.282 | 174,305 | 47.7 |
| FRANKENSTEIN | 5 / 6 | 1N | 77.703±1.604 | 186,977 | 29.5 |
| FRANKENSTEIN | 6 / 6 | 1N | 77.772±1.378 | 199,649 | 52.9 |
| FRANKENSTEIN | 3 / 6 | 1NC | 77.957±1.972 | 192,961 | 47.9 |
| FRANKENSTEIN | 4 / 6 | 1NC | 77.426±1.219 | 205,633 | 29.8 |
| FRANKENSTEIN | 5 / 6 | 1NC | 78.418±0.960 | 218,305 | 68.4 |
| FRANKENSTEIN | 6 / 6 | 1NC | 77.311±1.245 | 230,977 | 72.7 |
| NCI1 | 3 / 6 | no | 86.083±1.747 | 95,233 | 55.9 |
| NCI1 | 4 / 6 | no | 86.010±1.762 | 107,905 | 62.2 |
| NCI1 | 5 / 6 | no | 86.107±1.623 | 120,577 | 66.2 |
| NCI1 | 6 / 6 | no | 85.961±2.009 | 133,249 | 63.5 |
| NCI1 | 3 / 6 | 1N | 85.669±1.677 | 114,017 | 53.2 |
| NCI1 | 4 / 6 | 1N | 86.326±1.635 | 126,689 | 67.9 |
| NCI1 | 5 / 6 | 1N | 86.642±1.739 | 139,361 | 78.2 |
| NCI1 | 6 / 6 | 1N | 86.375±1.775 | 152,033 | 66.2 |
| NCI1 | 3 / 6 | 1NC | 85.401±1.419 | 145,345 | 50.0 |
| NCI1 | 4 / 6 | 1NC | 86.010±1.687 | 158,017 | 70.0 |
| NCI1 | 5 / 6 | 1NC | 86.375±1.592 | 170,689 | 93.7 |
| NCI1 | 6 / 6 | 1NC | 86.204±1.788 | 183,361 | 74.2 |

We conduct ablation study and hyper-parameters tuning experiments for ERFGN$^\circ$ and ERFGN$^\circ$+GloAttNC on TUDatasets (Morris et al., 2020a), separately. The function is CrossEntropyLoss, the optimizer is AdamW (Loshchilov & Hutter, 2018), the batch size is 64, and the learning rate is $10^{-3}$, halved every 50 epochs before falling below $10^{-5}$. The maximum number of

epochs is 150, but training will stop early after 100 epochs if performance on the validation set does not improve. Other experimental setup follows those of (Xu et al., 2019). The results are shown in Table 17.

