# OpenReview forum: "Efficient Redundancy-Free Graph Networks: Higher Expressiveness and Less Over-Squashing"
_ICLR.cc/2024/Conference — Submitted to ICLR 2024_

### Official Review · Reviewer_vob1 · 2023-10-26

**Soundness:** 3 good
**Presentation:** 1 poor
**Contribution:** 2 fair
**Rating:** 5
**Confidence:** 3

**Summary:**

This paper points out that the message passing mechanism entails redundancy, limiting expressiveness and cause over-squashing. To solve this, the authors presents solutions based on directed line graph / directed acyclic line graph. Furthermore, they show that the solution improves expressiveness, and validate it across various benchmarks.

**Strengths:**

1. The paper points out the cause of redundancy in graph neural networks into two aspects, cycles and backtracking, and eliminate it by using path trees and directed line graphs (DLG).
2. A theoretical analysis on the expressive power exists for the proposed method, though under certain conditions.

**Weaknesses:**

1. The paper title says “higher expressiveness” and “less over-squashing”. While there exist a theoretical analysis for the expressive power in section 2.7, there are no explanation or mention about over-squashing neither in theory or experiments.
2. The paper doesn't have any expressive power analysis for conventional GNNs, while they do for DLGN and ERFGN. When using k-WL for the expressivity power of GNNs for example, one claims that a new architecture is as powerful as the 2-WL test while conventional GNNs are as powerful as the 1-WL test$^{[1]}$, concluding that the new architecture is more powerful than the original. However, the paper lacks expressive power analysis for conventional GNNs with ${L}$ layers, and only propose the expressive power analysis for DLGN and ERFGN.
3. The paper is somewhat difficult to read, with some typos (ex. For table 4, there are only 2 datasets while the caption states three datasets, while for table 5, they are no bold results for the ogbg-molhiv dataset).

[1] Xu et al., How Powerful are Graph Neural Networks?, ICLR 2019

**Questions:**

1. Following weakness #1, are there any theoretical and experimental support for the claim DLGN/ERFGN results less over-squashing?
2. I have been familiar to using the term expressive power of graph neural networks to measure the ability of the model distinguishing non-isomorphic graphs. How can the graph radius used to be to measure the expressive power of graph neural networks?
3. Following Table 1, authors claim that DLGN and ERFGN shows improved efficiency in addressing message passing redundancy. However, the complexity of DLGN/ERFGN seems to be higher when compared to the complexity of typical MPNNs. Does the author mean the proposed method offers improved efficiency compared to the prior work, DLGN? If so, it would have been clearer by adding a checklist row at the bottom of Table 1, whether the method considers redundancy.

---

> ### Author Response · Authors · 2023-11-20
>
> We express our sincere gratitude to the reviewer for their constructive review. We highly value their feedback and have diligently addressed their concerns by extensively revising the entire manuscript to ensure that the manuscript meets the standards of a qualified and reader-friendly publication.
>
> We ask the reviewers for their patience and kindness in dealing with this revised manuscript.
>
> [W1]
>
> We have already discussed how our models can reduce the over-squashing problem in Appendix C of our previous manuscript. That message passing with redundancy can lead to the over-squashing problem was discussed in the Introduction section (in the middle of the first paragraph on page 2). Chen et al. [^1] show that a redundancy-free message passing scheme can ensure that only one message passing path captures each path, thus addressing the imbalance in the number of message passing paths generated from different paths and mitigating the over-squashing problem. For the same reason, our ERFGN model can mitigate the over-squashing problem. Our experiments demonstrate that DLGN outperforms typical MPNNs, higher order MPNNs, and graph transformers on MNIST, CIFAIR10, and Peptides-func. While DLGN cannot achieve higher expressiveness because it doesn't address the message confusion problem caused by native cycles, it still benefits from the reduced over-squashing problem. These experimental results support the advantages of mitigating over-squashing.
>
> We agree that it is important to include a clear explanation of how to reduce over-squashing. We incorporated a detailed explanation of how our models mitigate over-squashing in Section2.7 of the revised manuscript. Additionally, we discussed the benefits of mitigating over-squashing in the Experiments section (experiments on MNIST, CIFAR10, Peptides-func and Peptides-struc).
>
> [W2]
>
> We appreciate the reviewer's comment on the comparison with conventional GNNs using k-WL. In the revised version of our manuscript, we have included a lemma in Section 2.6 that highlights the increased power of our model compared to the 3-WL test. We have also provided proof in the appendix (on page 21) to support this claim.
>
> [W3]
>
> We appreciate the feedback and have diligently rewritten the entire manuscript to ensure its correctness and clarity.
>
> [Q1]
>
> In the revised manuscript, we have included a detailed explanation of how our models mitigate over-squashing in Section2.7 and discussed the benefits of mitigating over-squashing in the Experiments section.
>
> [Q2]
>
> We apologize for the confusion caused by our incorrect presentation.  Actually, the graph radius determines the minimum number of message passing iterations required to transfer information from all nodes in a connected graph to a certain node and to express all paths of the connected graph. It also determines the minimum number of layers of DLGN or ERFGN to achieve their full potential expressiveness, respectively. We have corrected this error in our revised manuscript.
>
> [Q3]
>
> We aim to propose an efficient method to achieve redundancy-free message passing. It has been discussed in Introduction (first paragraph on page 2) that RFGNN[1] is the first previous work that achieves redundancy-free, but it faces complexity challenges. Our proposed methods offer improved efficiency compared to RFGNN.
>
> We have added a checklist row to Table 1 to indicate whether or how well a method can address redundancy. In addition, we have made our motivation and contributions more clear in the revised manuscript.
>
> [^1]: Rongqin Chen, Shenghui Zhang, Leong Hou U, and Ye Li. Redundancy-Free Message Passing for Graph Neural Networks. Advances in Neural Information Processing Systems, NeurIPS, 35: 4316–4327, 2022.

---

> > ### Comment · Reviewer_vob1 · 2023-11-23
> >
> > I thank the authors for their comprehensive response, raised my score, and have no further questions. The authors could improve their paper by the following.
> >
> > ---
> >
> > ### W1/Q1
> >
> > **Mitigation of over-squashing**
> >
> > The authors have added an explanation of “less Over-squashing” in section 2.7. The first paragraph explains the related works, and the second paragraph states that ERFGN follows the same redundancy-free message-passing strategy as RFGNN. Unfortunately, the paper still misses a formal theoretic explanation related to the mitigation of over-squashing, such as using the sensitivity, access time, or any mathematical aspects aligning with prior works $^{[1,2]}$.
> >
> > The authors have just implied that “ERFGN exhibits a similar sensitivity to capturing subtree as RFGNN”, and rather than this, I suggest adding an appendix section with a formal description of the commonalities and differences with RFGNN. With the "relative influence of trail", it would support the claim of mitigating over-squashing. The connection between the two methods doesn't seem hard.
> >
> > **Efficiency from experiments**
> >
> > In the updated section 3: experiments, authors claimed that the proposed models are efficient in presenting the “per-epoch/total training time” for dataset MNIST, CIFAR, Peptides-func, and peptides-struct. To show this more effectively, I think they should add “per-epoch/total training time” of previous works & baselines neither in the main paper nor the appendix. To say that ERFGN is efficient requires a baseline, i.e., the practical time of previous works.
> >
> > ---
> >
> > ### W2
> >
> > The authors have added a lemma in Section 2.6 and Appendix C.5 that highlights the increased power of our model compared to the 3-WL test. Though this was not my question exactly. What I suggested was that to show the strength of ERFGNs, i.e., capable of expressing subtrees of non-isomorphic graphs, it would been better to add a lemma of ordinary/conventional GNNs failing to express all subtrees of non-isomorphic graphs. For instance, it would be like “1-WL-GNNs, i.e., GNNs having expressiveness equivalent to 1-WL test, with L layers cannot express all subtrees of non-isomorphic graphs in some conditions”.
> >
> > Although it was not intended, the authors have added a comparison with the WL test, which is quite interesting. However, it would be better if the authors could explain the lemmas and proofs more formally, and add a short explanation about lemmas 5~8 in section 2.8.
> >
> > 1. Lemma 5
> >
> > Though one can easily assume what a graph component is, i.e., a connected subgraph that is not part of any larger connected subgraphs, the formal definition is not given. It would be better if it was written in somehow.
> > Also, rephrasing the lemma such as “ERRFGN can distinguish every graph component that contains a cycle, while 1-WL-GNN cannot” seems more strong. To show that ERFGN is capable of doing something, once again, it would be more persuasive to show that conventional GNNs (1-WL-GNN) cannot do it. This would be a simple proof, I assume.
> >
> > 2. Lemma 8
> >
> > The formal definition of subgraph expressiveness seems to be missing. Though the authors stated “expressiveness of GNNs” in the section 1 introduction, subgraph expressiveness needs to be formally defined.
> > Also, the proof of Lemma 8 lacks details. To say that ERFGN is strictly more powerful than 3-WL-GNNs, one usually to show two facts in prior works (Appendix D.2.1, Proof of theorem 1 in Bevilacqua et al.$^{[3]}$ would be a good example).
> >
> > (1) All graph pairs distinguishable by 3-WL-GNN are also distinguishable by ERFGN
> >
> > (2) There exists a graph pair distinguishable by ERFGN, but undistinguishable by 3-WL-GNN
> >
> > Authors have implicitly shown (2) using Bodnar et al. (2021), “3-WL cannot count chord-less cycles of size strictly larger than 3”, while ERFGN can. However, the authors have not shown (1), showing this would make a clear and complete proof.
> >
> >
> > &nbsp;
> >
> > &nbsp;
> >
> > [1] Topping et al., understanding over-squashing and bottlenecks on graphs via curvature
> >
> > [2] Black et al, Understanding oversquashing in GNNS through the lens of effective resistance, ICML 2023
> >
> > [3] Bevilacqua et al., Equivariant subgraph aggregation networks, ICLR 2022

---

> > > ### Author Response · Authors · 2023-11-23
> > >
> > > We thank the reviewer for their constructive comments.
> > >
> > > Unfortunately, we don't have time to do an update of our manuscript before the end of the rebuttal, but we'll do our best to improve our work anyway.

---

### Official Review · Reviewer_WgZE · 2023-10-31

**Soundness:** 3 good
**Presentation:** 3 good
**Contribution:** 3 good
**Rating:** 6
**Confidence:** 3

**Summary:**

Two surrogate structures for graphs are proposed: Directed Line Graphs (DLGs) and Directed Acyclic Line Graphs (DALG). They provably enable relatively flexible message passing neural networks (DLGN and ERFGN) to distinguish non-isomorphic graphs. This makes them provably expressive in graph classification tasks. They further allow to achieve redundancy-free message passing to address over-squashing.

**Strengths:**

- Directed Line Graphs (DLGs) are proposed as surrogate structure of graphs to enable non-backtracking message passing. This is supposed to overcome issues with over-squashing, message confusion, and expressiveness of GNNs.
- The composition of multiple algorithms are proposed to convert a graph into a DLG or DALG.
- The expressive power of the proposed GNN architectures is proven by showing that they could distinguish non-isomorphic graphs.
- The runtime of the proposed graph conversion and message passing models are analysed. Even though the runtime is considerable (see weaknesses), at least the message passing scheme of DLGN is more efficient than the typical one (RFGNN) that also tries to avoid redundant messages.

**Weaknesses:**

- Alternative approaches to reduce message redundancy (like SPAGAN (Yang et al., 2019) or PathNNs (Michel et al., 2023)) could be discussed in more detail and be compared to in experiments.
- The proof of Lemma 1 (as well as Lemma 2) is not detailed and does not verify that each step of the conversion actually defines a bijection. In this sense, the proof is not complete.
- The proposed graph conversion is computationally very costly and therefore does not scale to large graphs. In particular, the TPT extraction and sub-DALG construction have a complexity of $O(|V_C|!)$.
- Furthermore, dense graphs would have large DALGs.
- The definition of the proposed architectures (e.g. ReadNCG) require training relatively complex models with a high number of trainable parameters and trainable functions that might be difficult to train in practice.
- Many of the experimental evidence that is presented does not lead to significant improvements.

Minor points:
- Chordless cycles are not introduced. Their knowledge is just assumed.
- It could help the reader to present an example where backtracking messages actually present a problem for the expressiveness of GNNs.

**Questions:**

- Does the removal of circles not imply that a graph with cycles and the same graph without cycles become indistinguishable?
With additional message aggregation for cycles, this is addressed, but how does the aggregator need to look like so that full expressiveness is achieved? Can this aggregator be learned in practice?
- Please add a precise definition of ERFGN.
- Please add a definition of the maximum radius of a component of DLGN or ERFGN.
- Is it clear that backtracking messages are bad for solving a task? Couldn't they also cover computations that are not realisable by DALG or ERFGN?

---

> ### Author Response · Authors · 2023-11-20
>
> [W1]
>
> We sincerely appreciate the reviewer's suggestion to discuss and compare alternative approaches for reducing message redundancy. We have included a brief discussion and a comprehensive discussion of alternative approaches in Introduction (2nd paragraph of page 2) and in the related works section (Appendix A). This addition strengthens the context and significance of our research.
>
> The experimental results of PathNNs have been included in our experimental section and compared with other methods. However, SPAGAN focuses on node classification tasks, which are quite different from ours, i.e., graph-level predictions. And their codes could not be used directly for graph-level predictions. Therefore, SPAGAN is not compared in our experiments.
>
> [W2]
>
> We have thoroughly revised and expanded the proofs in Appendix C.2 of our revised paper to ensure they are detailed and complete.
>
> [W3]
>
> The complexity analysis we performed considered  the worst-case scenario, assuming that the cyclic subgraph is a complete graph. So $|V_C|$ path trees are extracted, each root node has $|V_C|-1$ child nodes, and each child node has $(|V_C|-2)$ 2-order child nodes, and so on.
>
> However, real-world graphs, such as molecules, are often sparse and consist of small, biconnected components, which would not result in very high complexity.And we have drawn a more precise analysis: Let $V^{\circ}$ be the nodes of the cyclic subgraph, and $d^{\circ}$ be the maximum node degree. The extracted path trees consist of $|V^{\circ}| \times d^{\circ}!$ nodes and edges at worst. Therefore, $|V^{\circ}|$ path trees are extracted, where each root node at worst has $d^{\circ}$ child nodes, and each child node has $(d^{\circ}-1)$ 2-order child nodes, and so on. Thus, for ERFGN, the path tree extraction, as well as the sub-DALG construction, has a complexity of $\mathcal{O}\left(|E|+|V^{\circ}| \times d^{\circ}!\right)$.
>
> [W4]
>
> We appreciate the question regarding our method's suitability for dense graphs. To address this, we are exploring adaptations such as graph decomposition. Following the success of ESAN[^1], we can decompose a dense graph into multiple spanning trees and repeat the process with new spanning trees to obtain topological properties while handling dense graph complexities. We would like to investigate this problem further and welcome any suggestion from the reviewer.
>
> [W5]
>
> Our proposed architectures exhibit complexity due to the need to model global interactions between different pairs of nodes and cycles. To address this complexity, we have developed two global attention schemes: a 1-to-N scheme and a 1-to-1 scheme. These schemes cater to different priorities, either prioritizing efficiency or fine-grained learning.
>
> We have conducted extensive empirical evaluations on diverse datasets to demonstrate the trainability and effectiveness of the proposed schemes. Our experiments show that with appropriate training procedures and hyperparameter tuning, the 1-to-N scheme achieves high efficiency and desirable performance levels on many datasets, while the 1-to-1 scheme also achieves desirable performance levels on some datasets. Detailed experiment results can be found in Appendix E of the revised manuscript.
>
> [W6]
>
> To further support the effectiveness of our model, we have evaluated our methods on a new dataset called QM9, which has 12 tasks, and the experimental results show that our model achieved significantly better results than the benchmark method in multiple tasks.
>
> [Minor point 1]
>
> We have introduced this concept of 'chordless cycles' in Section 2.1 of the revised manuscript.
>
> [Minor point 2]
> In the 2nd paragraph of the Introduction in the revised manuscript, we present an example where backtracking messages cause message passing redundancy and show that message passing redundancy causes the message confusion problem.

---

> > ### Author Response · Authors · 2023-11-20
> >
> > [Q1]
> >
> > Yes, removing the circles would make a subgraph with cycles indistinguishable from a subtree. However, representing a cycle as an epath also makes a subgraph with cycles indistinguishable from a subtree with one more hop neighbor. This is because it is not known whether an epath represents a cycle or just a path. That is, explicitly modeling subcycles is a better way to distinguish between a subgraph and a subtree. By the way, the removal of circles will not affect the distinguish of whole graphs because there is at least one subtree in one graph but not in the non-isomorphic graph.
> >
> > Since the subtrees extracted from the nodes in a cycle retain all the topological information about the cycle and the subgraph containing the cycle, it is possible to represent a cycle with the subtrees extracted from the nodes in the cycle. A graph readout, which aggregates the information of all subtrees extracted from a graph, has proven to be able to express non-isomporhic graphs. Therefore, a cycle aggregator could be the same as a graph readout, such as a summation function, to achieve the full expressiveness of cycles. And the trainability of cycle aggregators can be proved in the same way as the trainability of graph readouts.
> >
> > We sincerely thank the reviewer for pointing out the conditions for achieving an expressive cycle aggregator. We have added a discussion of the condition in the proof of the expressiveness of ERFGN$^{\circ}$  (i.e., Proof of Lemma 5 on page 23)
> >
> > [Q2] We have made the definition of ERFGN precise in Section 2.4 of the revised manuscript.
> >
> > [Q3] Actually, it is the maximum radius of the graph components. We have carefully rewritten these definitions.
> >
> > [Q4] For solving the expression of non-isomorphic subgraphs, backtracking messages are bad. This is because backtracking messages do not lead to new information about the topoloy, but they do cause confusion and over-squashing. For other tasks, it is possible that backtracking messages cover computations that are not realizable by DALG or ERFGN.
> >
> > [^1]: Beatrice Bevilacqua, Fabrizio Frasca, Derek Lim, Balasubramaniam Srinivasan, Chen Cai, Gopinath Balamurugan, Michael M. Bronstein, and Haggai Maron. Equivariant Subgraph Aggregation Networks. International Conference on Learning Representations, ICLR, 2022.

---

> > > ### Comment · Reviewer_WgZE · 2023-11-22
> > >
> > > I thank the authors for their detailed response. I do not have any further questions.
> > >
> > > According to the discussion with the other reviewers, the theoretical contribution of this work seems to be limited, yet, I appreciate the improvement in run time over competing methods. However, the run times of the proposed method will likely hinder scaling the proposed approach to graphs even of medium size. For that reason, I have reduced by score to 6. However, I will remain open to adapting my score during the discussion of the reviewers.

---

> ### Author Response · Authors · 2023-11-23
>
> Dear reviewer,
>
> We thank you for your kind feedback.
>
> We hope that you will be able to review the contribution of our work and the practical efficiency of our approach as summarised in our public response.

---

### Official Review · Reviewer_Gehp · 2023-11-01

**Soundness:** 2 fair
**Presentation:** 2 fair
**Contribution:** 2 fair
**Rating:** 3
**Confidence:** 4

**Summary:**

The authors propose to transform the input graphs of GNNs into directed (acyclic) line graphs and process them using custom neural architectures. The transformations aim at reducing redundancy in message passing. Experimental results show improvements of the proposed method.

**Strengths:**

1. Redundancy is a key problem in message-passing GNNs; reducing it has been shown to alleviate oversquashing.
2. The approach builds on recent work in the same direction.
3. Experimental results are promising.

**Weaknesses:**

1. The presentation is not sufficiently clear and several claims and results require further substantiation:
   - The construction of the DALG is fundamental for the work, but its description (list with 7 steps on page 4) needs to be clearer: In step 1 the chordless cycles are extracted from the graph. While not mentioned in the paper, this can be an extremely expensive step as the number of chordless cycles can be exponential in the graph size. However, the chordless cycles are only used in step 2 to partition the graph into two components based on the edges contained in cycles and those that are not contained in cycles. However, this can be achieved in linear time using a standard algorithm for finding biconnected components. The crucial part of the construction is the generation of path trees for the biconnected components using an approach closely related to RFGNN proposed by Chen et al. (NeurIPS 2022). These are finally combined with the representation of the acyclic part. No motivation is given as to why it is advantageous to build the path trees only for the biconnected components.
   - The complexity analysis needs to be clarified: While it is claimed that the approach is efficient, this is not clear from the construction of the transformed graphs and the analysis in Section 2.8. First, I would expect no advantage of the approach on biconnected graphs compared to RFGNN. Unfortunately, the parameter $V_C$ used to specify the running time has not been explained. If $V_C$ is the set of nodes in the biconnected components (which I assume), I do not understand why the approach is considered efficient, since its running time is factorial in $|V_C|$. If it is only efficient for graphs with small biconnected components (like molecular graphs) this limitation needs to be made explicit.
   - Minor remarks:
      - In the introduction, walks are specified by their label sequence. It would be much clearer if vertex identities were used instead.
      - Section 2.1: The sentence "A cycle $c \in C$ consists of connected nodes/edges in graph $G$." is not a definition of cycles; and also not a helpful statement.
      - Section 3: The conclusion "The experimental results demonstrate the expressive power of our models." is not justified. As the results show test accuracy (I assume), they also reflect the generalization of the approach.
2. The contribution is incremental and the novelty is limited.
   - A similar path tree has been proposed for RFGNN by Chen et al. (NeurIPS 2022).
   - Directed line graphs have been used in several other papers, e.g.,
      - Pierre Mahé, Nobuhisa Ueda, Tatsuya Akutsu, Jean-Luc Perret, Jean-Philippe Vert: Extensions of marginalized graph kernels. ICML 2004
      - Zhengdao Chen, Lisha Li, Joan Bruna: Supervised Community Detection with Line Graph Neural Networks. ICLR 2019

      The first paper by Mahé et al. introduced the idea of avoiding backtracking to graph learning, more specifically, to random walk kernels. The second paper introduced a similar technique for GNNs. Both papers are not cited.  The novelty of the specific use of (directed) line graphs needs further discussion.

3. The expressivity analysis is not sufficiently rigorous. First, there are statements that need justification, e.g., the proposed approach is claimed to achieve "higher-order expressiveness" (abstract). However, the method is not formally related to k-WL (if the term refers to this). I have some concerns regarding the proof of Lemmas 1 and 2 (in the appendix): The "if and only if" statements require to show two directions. Unfortunately, only one direction is discussed in detail; namely, that isomorphic graphs lead to isomorphic transformed graphs. However, the reverse direction is much more interesting. These are not discussed but just claimed to be true. A rigorous proof is necessary.

**Questions:**

1. Is my understanding that the cyclic subgraph is the union of all biconnected components?

---

> ### Author Response · Authors · 2023-11-20
>
> [W]: We deeply apologize for the unclear presentation in the previous version. We have taken this feedback seriously and have carefully rewritten the entire manuscript to ensure that it meets the standards of a qualified and reader-friendly publication.
>
> We kindly ask reviewers to approach the new version with patience and kindness, and we sincerely hope that our efforts have resulted in a truly readable manuscript. Thank you for your understanding and consideration.
>
> [W1.1]:
>
> (clarity): We have made the necessary revisions to clarify the steps involved in the revised manuscript.
>
> (cycle extraction): The complexity of extracting cycles was given in Section 2.8 of the previous version. It is important to note that the purpose of extracting cycles extends beyond graph splitting. The extracted cycles also play a crucial role in cycle modeling, which justifies the necessity of the cycle extraction process. Regarding the time cost of constructing DLGs and DALGs, we conducted experiments on more than 10 datasets. The experiment results presented in Appendix E of our revised manuscript demonstrate that these procedures are efficient.
>
> (relation to RFGNN): RFGNN (Chen et al.) extracts a path tree from each node, so it suffers from a complexity problem. Before our work, there was no solution that allowed efficient redundancy-free message passing. We proposed efficient redundancy-free message passing by extracting path trees rather than epath trees from cyclic subgraphs rather than whole graphs. Our approach is different from that of RFGNN, and ours is more efficient and scalable.
>
> (motivation):
>
> Our revision to the motivation is summarized as follows:
>
> - Redundancy-free message passing is of the utmost importance. RFGNN achieves this goal. However, it suffers from computational and memory complexity, which limits its scalability and applicability. We found that message passing redundancy can be addressed by eliminating cycles. Additionally, extracting path trees only from cyclic subgraphs offers efficiency and scalability over extracting path trees from entire graphs. Therefore, we propose our methods, which only transform the cyclic subgraphs into cycle-free surrogate subgraphs.
>
> - Furthermore, to capture higher-order graph structures like cycle-based subgraphs, we propose two extensions to the ERFGN framework. The first is ERFGN$^{\circ}$, which incorporates a cycle modeling module to represent any subgraph with a cycle. The second extension is ERFGN$^{\circ}$+GloAttNC, which introduces global attentions between nodes and cycles. This enhancement enables the expression of subgraphs consisting of subtrees and subcycles, and facilitates the generation of identifiable representations for nodes and cycles.
>
> [W1.2]:
>
> (complexity):
>
> - The complexity analysis has been updated with clearer and more precise information in Section 2.8. In addition, empirical time costs for constructing the DLGs and DALGs are provided in Appendix E. We have also clarified the limitation of our methods to be suitable for real-world graphs (such as molecules), which are sparse and have small cyclic subgraphs. Compared to RFGNN [^1], which is the first work that achieves the redundancy-free feature, our methods offer obvious efficiency and higher expressiveness.
>
> - While our methods may not have a significant advantage over RFGNN for biconnected graphs, it is worth noting that there exist real-world graphs, such as molecules, which are often sparse and consist of small, biconnected components. Empirically, our methods have shown efficiency across over 10 datasets (see Appendix F), indicating their suitability for real-world scenarios.
>
> (parameter $V_C$):
>
> We have explained the parameter $V_C$ in Section 2.8 of the revised manuscript.  Previously, our complexity analysis was based on the worst-case scenario where the cyclic subgraph is a complete graph. However, this scenario is not close to reality. We have revised the analysis to make it more precise. The revised analysis is summarized as belows:
>
> Let $V^{\circ}$ be the node set of the cyclic subgraph, and $d^{\circ}$ be the maximum node degree. The extracted path trees consist approximately of $|V^{\circ}| \times d^{L}$ nodes and edges. Therefore, for ERFGN, the path tree extraction, as well as the sub-DALG construction, has a complexity of $\mathcal{O}\left(|E|+|V^{\circ}| \times d^{L}\right)$ at worst.

---

> ### Author Response · Authors · 2023-11-20
>
> [W1.3 Minor remarks]
>
> - We represent walks in our study using their label sequences because typical MPNNs construct message passing graphs based on node labels and connections without taking into account the node identities. By denoting walks with their label sequences, we aim to provide a clear understanding of how "message confusion" can occur. In addition, we have made efforts to enhance the clarity of our explanations in the revised manuscript.
>
> - We have rectified the issue by replacing the incorrect sentence with a correct definition of "chordless cycle" in the revised manuscript.
>
> - Test accuracy alone may not fully capture the expressiveness of an approach. As a result, we have made new conclusions in our revised manuscript:
>
>     1. Our models demonstrate higher expressiveness by effectively representing a wide range of subgraphs, including subtrees, subgraphs with cycles, and subgraphs with multiple cycles.
>
>     2. Empirical results strongly support the efficiency and high performance of our models.
>
> In addition, it is worth noting that the high expressiveness of the model is essential for achieving better generalization performance. An ideal approach should effectively capture a wide range of non-isomorphic subgraphs to accurately measure graph similarity and predict graph properties [^2]. Conversely, methods lacking sufficient expressiveness for non-isomorphic subgraphs often struggle to measure graph similarity. While such methods may effectively distinguish training graphs, they lack the ability to accurately measure graph similarity, leading to overfitting. Our experimental results fully support this conclusion, demonstrating that our models (ERFGN, ERFGN$^{\circ}$, and ERFGN$^{\circ}$+GlobalAttention) achieve near-perfect performance on training datasets and exhibit improved generalization with higher expressiveness.
>
> [W2]:
>
> Despite the similarities between our method and RFGNN, the differences between us are not negligible. Our method only extracts the path tree rather than the epath tree from the cyclic subgraph rather than the full graph, which makes our method significantly more efficient compared to RFGNN.
>
> We thank the reviewer for providing us with relevant work that we had overlooked. The discussion about these works has been added to the 2nd & 3rd paragraphs on page 2 of the revised manuscript.
>
> Our DLG structure is similar to those directed line graphs used in (Mahé et al; Chen et al.), but with differences. First, the structure used in (Mahé et al.) has a larger size than ours. A directed line graph used in (Mahé et al.) constructs its own node set from the nodes and edges of its source graph. However, our DLG constructs its own node set only from the edges of its source graph. Small-scale surrogate graphs allow for efficient message passing. Therefore, using our DLG structure enables higher efficiency than that used in (Mahé et al.).
>
> In addition, a directed line graph used in (Chen et al.) also constructs its own node set only from the edges of its source graph. However, such a structure is not used to update the node representation. To update node representation, we add additional connections to DLG to line DLG nodes to source nodes, so that message of DLG can be conveyed to source nodes and it is possible to learn source subtrees from DLGs.
>
> We would like to emphasize that the contributions of our study extend beyond the scope of extracting path trees and utilizing directed line graphs. Our study encompasses broader contributions, which are summarized on page 3 of the revised manuscript.
>
> [W3]: We have justified our statements and added a comparison with GNNs using $k$-WL in Section 2.6. We have also carefully rewritten our proofs and proved the double directions of  each statement in Lemmas 1 and 2 to ensure their clarity and rigor in Appendix C.2.
>
> [Q1]: Yes, the cyclic subgraph is the union of all biconnected components. We appreciate again the reviewer's suggestion for a linear complexity method for extracting cyclic subgraphs. However, in our approach, we not only partition graphs but also extract cycles to model the cycle-based subgraphs. Therefore, we adopt the process of extracting the cycles and then constructing the cyclic subgraph.
>
> [^1]: Rongqin Chen, Shenghui Zhang, Leong Hou U, and Ye Li. Redundancy-Free Message Passing for Graph Neural Networks. Advances in Neural Information Processing Systems, NeurIPS, 35:4316–4327, 2022.
>
> [^2]: Anton Tsitsulin, Davide Mottin, Panagiotis Karras, Alexander M. Bronstein, and Emmanuel Muller. NetLSD: Hearing the shape of a graph. ACM SIGKDD Conference on Knowledge Discovery and Data Mining, KDD, 2018.

---

### Official Review · Reviewer_Dnf7 · 2023-11-04

**Soundness:** 3 good
**Presentation:** 3 good
**Contribution:** 3 good
**Rating:** 6
**Confidence:** 3

**Summary:**

This paper discusses improved heuristics for the ``redundancy-free'' design of graph neural networks. The development of the GNN leverages the line graph and makes substantial improvements in computational complexity by mining the acyclic line graph structure. An extended model for the interplay between DALG and cycles is also proposed.

**Strengths:**

1. This paper is clearly motivated and nicely organized.
2. The empirical improvement is significant.

**Weaknesses:**

Given that the major motivation and contribution of this paper are easy to follow, I suggest that further details should be provided to help readers further understand this method from theoretical and practical aspects, please check my questions.

**Questions:**

1. What are the empirical time costs of constructing the DALG for DLGN / ERFGN for different datasets?
2. Given that DLGN follows the message-passing scheme on DLG/DALG and the isomorphism of DLGs/DALGs is equivalent to the isomorphism of the original graphs, can we conclude the expressiveness of DLGN is upper bounded by 1-WL test? Can the interplay of DALG and cycle improve the expressiveness?
3. Table 6 and 7 are nice, it could be even better if the authors would like to compare the comparison of #parameters for maybe the most competitive baselines.
4. What is the empirical effect of changing tree-height $L$? How to choose a proper number of ERFGN layers if L changes?

---

> ### Author Response · Authors · 2023-11-20
>
> [W]:
> We appreciate the reviewer's feedback and have made thorough revisions to address their comments. The manuscript now provides comprehensive details on both the theoretical and practical aspects of our work. In particular, expressiveness is discussed and compared with other powerful methods in Section 2.7 and proved in Appendix E. Complexity is analyzed in Section 2.8. Empirical results, including prediction performance and efficiency, are given in Section 3 and detailed in Appendix D.
>
> We kindly request that the reviewers approach the new version with patience and kindness. We appreciate your understanding and consideration. In addition, to avoid possible misunderstandings, we would like to point out that DLGN conducts message passing on DLGs while ERFGN conducts message passing on DALGs.
>
> [Q1]:
> The empirical time costs of constructing the DLGs and DALGs for different datasets are listed in tables in Appendix E of the revised paper. Please refer to Appendix E for a comprehensive overview of the empirical time costs.
>
> [Q2]:
> We apologise for the unclear presentation. It is true that DLGN's expressiveness is limited by the 1-WL test due to the message confusion problem caused by cycles. However, DLGN can still outperform 1-WL based GNNs by reducing message passing redundancy and mitigating the over-squashing problem. In addition, we have added an expressiveness ranking of our models and other powerful models in Section 2.6 of the revision paper.
>
> Yes, the interplay of DLGs and cycles can enhance expressiveness. This interplay allows a model to achieve the same level of expressiveness as CIN (Cell Isomorphism Network) [^1], which surpasses the 3-WL test. However, it is worthy to note that such a model suffers from the message redundancy problem, as well as the message confusion and message over-squeezing problems associated with message redundancy.
>
> [Q3]:
> Actually, the number of parameters will not vary much from model to model; this is because all models follow a parameter budget on most datasets. We have included parameter budgets in Section 3.
>
> [Q4]:
> The effect of changing tree-height $L$ has been refined in Tables 12, 13 and 17 in Appendix F.
> In our experiments, increasing $L$ improves performance, but with diminishing returns. Different datasets require different $L$. For example, long-range interaction datasets require large $L$. However, for small molecular datasets, a small value of $L$ is sufficient. In addition, $L$ not need to exceed the radius of the largest graph component. When $L$ is set equal to this maximum radius, our model reaches its full potential expressiveness, allowing it to represent all possible subtrees and subgraphs in the dataset.
>
> [^1]: Cristian Bodnar, Fabrizio Frasca, Nina Otter, Yuguang Wang, Pietro Lio, Guido F. Mont´ufar, and Michael M. Bronstein. Weisfeiler and Lehman Go Cellular: CW Networks. Advances in Neural Information Processing Systems, NeurIPS, 34:2625–2640, 2021a.

---

### Author Response · Authors · 2023-11-23

Dear reviewers,

We would like to emphasize the contributions of our work and provide some evidence to support the practical efficiency of our methods.
In addition, we have revised our paper again in order to further improve the presentation of the manuscript and the rigor of the proofs. We kindly request that the reviewers approach the new version with patience and kindness.

**Contribution**

1. Redundancy-free message passing is critical. Previous methods cannot achieve this goal efficiently and perfectly. We revealed the nature of redundancy in message passing, i.e. the repeated traversal of messages along cyclic structures within surrogate graphs. We then concluded that redundancy-free message passing can be achieved by using cycle-free surrogate graphs. However, previous studies had not considered this solution. Therefore, their methods either achieve the goal of redundancy-free computation by extracting path trees, or can only deal with message redundancy caused by backtracking. Based on the proposed cycle-free surrogate graphs, i.e. DALGs, we developed ERFGN, ERFGN$^{\circ}$ and ERFGN$^{\circ}$+GloAttNC with improved expressiveness, **where ERFGN$^{\circ}$ and ERFGN$^{\circ}$+GloAttNC provide strictly improved expressiveness compared to GNNs using the $3$-WL test.**

2. **Our ERFGN$^{\circ}$+GloAttNC model can effectively generate identifiable representations for nodes and cycles, which, to the best of our knowledge, cannot be achieved by existing methods.**

     - MPNNs with redundancy often suffer from message confusion problems. Therefore, MPNNs will generate identical message passing graphs for non-isomorphic subgraphs, and MPNNs face challenges in solving tasks involving inter-node distances.

     - Existing redundancy-free message passing networks cannot express subgraphs with cycles; they only express subtress.

     - Graph Transformers cannot effectively generate identifiable representations for nodes and cycles because the global attention mechnism requires nodes to be better identifiable within the graph and its substructures[^1]. For that reason, Graph Transformers rely on  good positional or structural encoding [^2].

     - Our ERFGN$^{\circ}$+GloAttNC can generate identifiable representations for nodes and cycles because it uses the representations of all subtrees and non-isomorphic (sub)graphs must contain non-isomorhic subtrees.

3. **We evaluated our approach on more than 10 datasets, demonstrated the high performance and efficiency of our model, and achieved SOTA performance on 9 datasets.**


**Efficiency**

The experimental results have confirmed that our method can efficiently compute tasks on realistic medium-sized graphs.

For example, the Peptides-func and Peptides-struct datasets each contain an average of 150.9 nodes and 307.3 edges. Our ERFGN$^{\circ}$+GloAttNC model can complete the entire training process in 10 minutes and achieve SOTA performance.

In addition, compared to Graph Transformers, which have gained popularity, our methods can offer higher efficiency. The GPS[^2] , a SOTA Graph Transformer with 4 layers, computes representations $4 \times N^{2} = 4 * 150^2 = 90K$ times on the Peptides-func dataset, where $N$ is the number of nodes. However, our ERFGN$^{\circ}$+GloAttNC model with 6 layers only computes representations $6 \times 569.2 + 150^2 \simeq 26K$ times, where 569.2 is the number of edges of the extracted DALGs of height 6. So our ERFGN$^{\circ}$+GloAttNC model is more efficient than graph transformers. In addition, our ERFGN$^{\circ}$+GloAttNC model can also achieve better results than SOTA graph transformers.



[^1]: Dwivedi, Vijay Prakash, and Xavier Bresson. "A generalization of transformer networks to graphs." arXiv preprint arXiv:2012.09699 (2020).

[^2]: Rampášek, L., Galkin, M., Dwivedi, V. P., Luu, A. T., Wolf, G., & Beaini, D. (2022). Recipe for a general, powerful, scalable graph transformer. Advances in Neural Information Processing Systems, 35, 14501-14515.

---

### Meta-Review · Area_Chair_Gq4P · 2023-12-13

**Metareview:**

This paper introduces a more efficient and less redundant message-passing framework for Graph Neural Networks (GNNs) to reduce their redundancy and improve their expressiveness. The empirical improvement reported in the paper are encouraging, and the paper is nicely organized.

On the other hand, the presentation of the paper can be improved to be more clear and self-contained. Closely related work (including RFGNN, SPAGN, and PathNNs) should be discussed in more detail. In the submission, the proof of Lemma 1 does not seem complete. Reviewers also feel that the theoretical contribution of this work seems to be limited.

**Justification For Why Not Higher Score:**

The presentation of the paper can be improved to be more clear and self-contained. Closely related work (including RFGNN, SPAGN, and PathNNs) should be discussed in more detail. In the submission, the proof of Lemma 1 does not seem complete. Reviewers also feel that the theoretical contribution of this work seems to be limited.

**Justification For Why Not Lower Score:**

The empirical improvement reported in the paper are encouraging, and the paper is nicely organized.

---

### Decision · Program_Chairs · 2024-01-16

Reject